# Collagen piezoelectricity in osteogenesis imperfecta and its role in intrafibrillar mineralization

Jinha Kwon[1] & Hanna Cho [1 ✉]

Intrafibrillar mineralization plays a critical role in attaining desired mechanical properties of bone. It is well known that amorphous calcium phosphate (ACP) infiltrates into the collagen through the gap regions, but its underlying driving force is not understood. Based on the authors' previous observations that a collagen fibril has higher piezoelectricity at gap regions, it was hypothesized that the piezoelectric heterogeneity of collagen helps ACP infiltration through the gap. To further examine this hypothesis, the collagen piezoelectricity of osteogenesis imperfecta (OI), known as brittle bone disease, is characterized by employing Piezoresponse Force Microscopy (PFM). The OI collagen reveals similar piezoelectricity between gap and overlap regions, implying that losing piezoelectric heterogeneity in OI collagen results in abnormal intrafibrillar mineralization and, accordingly, losing the benefit of mechanical heterogeneity from the fibrillar level. This finding suggests a perspective to explain the ACP infiltration, highlighting the physiological role of collagen piezoelectricity in intrafibrillar mineralization.

[1] Mechanical and Aerospace Engineering, The Ohio State University, 201W 19th Ave, Columbus, OH 43210, USA. ✉email: cho.867@osu.edu

From an engineering point of view, bone is a smart material that adjusts its mechanical properties in response to external loadings[1–3]. Understanding this unique characteristic of bone has great potential to yield vast gains in safety, effectiveness, and affordability of designs in diverse fields[4–6]. To achieve its adaptivity, bone actively remodels its internal architecture and composition by replacing old bone with new bone tissue. This newly formed bone matrix mainly consists of soft type I collagen, which is subsequently reinforced by accumulating amorphous calcium phosphate (ACP) mineral precursors from extracellular fluid[7–9]. Through this mineralization process, ordered mineral crystallites appear to form on the exterior of the collagen fibrils (extrafibrillar mineralization) and within the fibrils (intrafibrillar mineralization). Even though more than 70% of mineral contents are placed in the extrafibrillar space, the intrafibrillar mineralization plays a crucial role in attaining desired heterogeneity in mechanical properties of bone[10–12]. It is well known that net-charged ACP clusters are attracted and infiltrated into the intrafibrillar space through the gap regions of collagen fibrils and, then, transform into crystalized nanoplatelets and gradually grow along the fibrillar direction[13–15]. Although it seems that the crystallization of permeated ACP occurs thermodynamically without any chemical aids[16–18], a driving force for ACP attraction and infiltration through the collagen gap regions has not been fully clarified.

Given bone's ability to adapt itself to external loadings and ACP's net charge, we hypothesized that the piezoelectric behavior of a collagen fibril would help the ACP infiltrate into the intrafibrillar space in response to mechanical stresses. Although the piezoelectricity is an inherent property of a collagen fibril stemming from its non-centrosymmetric structure[19–21], the collagen piezoelectricity has not been spotlighted for the mineralization mechanism because of its unclear role in physiological conditions[22–24]. By contrast, other mechanisms, such as non-collagenous proteins (NCPs)[25–27], electrostatic interaction[28], and osmotic pressure[29], have been considered to account for the ACP infiltration, but they seem to fail to link bone mineralization with mechanical loading inputs. In addition, while streaming potentials surrounding a collagen matrix have been suggested to explain the stress generated potential (SGP)[23,30], they are also insufficient to unravel the reason why the ACP infiltration locally occurs at the gap regions. In our recent study based on Piezoresponse Force Microscopy (PFM), a single Type I collagen fibril revealed a repeated piezoelectric pattern, inversely correlated with fibril's periodic structure with D-spacing, displaying a higher piezoresponse at gap regions than overlap regions[31]. Taking into consideration that this periodic piezoelectric profile corresponds to the stiffness profile on a mineralized collagen fibril[32,33], the observed piezoelectric heterogeneity corroborated our hypothesis emphasizing the role of collagen piezoelectricity in intrafibrillar mineralization.

In this study, to further evaluate the hypothesis, we examined the piezoelectricity of collagen fibrils in Osteogenesis Imperfecta (OI), known as brittle bone disease, and compared it with the piezoelectricity of healthy collagen in wild type (WT) bone[34–36]. Figure 1 illustrates the hierarchical structure of WT and OI bones. OI bone has a genetic mutation in Type I collagen, which causes structural changes and abnormal properties such as a decrease of ultimate strength and fracture toughness as well as a higher Young's modulus at the whole bone level[37–39]. Many studies utilizing AFM (Atomic Force Microscopy), TEM (Transmission electron microscopy), and SAXS (Small-angle X-ray scattering) confirmed that OI bone undergoes abnormal mineralization, and the OI collagen loses its heterogeneity of mechanical stiffness along the fibrils from the fibrillar level[32,33,40–42]. If our hypothesis is correct, the irregular intrafibrillar mineralization of OI collagen

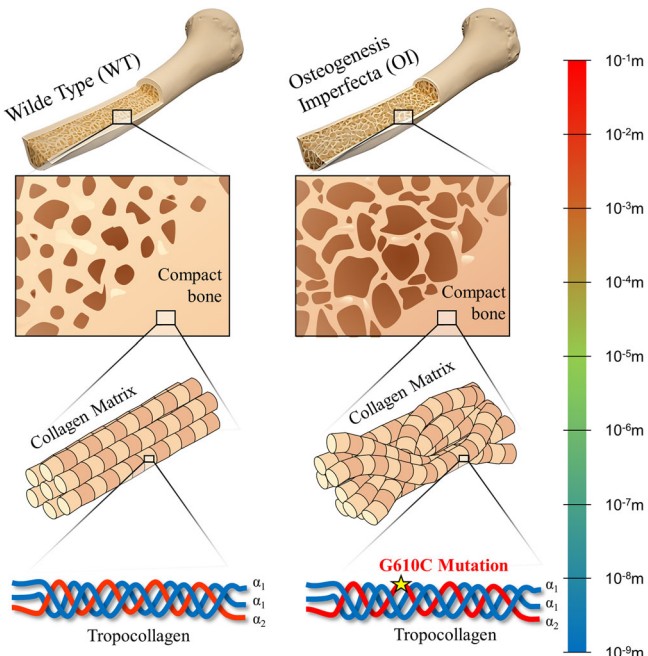

**Fig. 1 Hierarchical structure of WT and OI bones.** OI is a genetic bone disorder resulting in abnormal structure and material from the tissue to whole bone level, resulting in a decrease in ultimate strength and fracture toughness. The OI used in this study has the G610C mutation at the $\alpha_2$ chain of a tropocollagen. The OI collagen exhibits a slightly random orientation in its fiber-level matrix, while the WT collagen reveals a well-aligned structure. This multi-scale abnormality of OI results in a catastrophic effect on bone's mechanical integrity.

showing uniform stiffness distribution might be attributed to its defective piezoelectricity due to the mutation of collagen molecules. Thus, the OI collagen is a great platform to examine the role of collagen piezoelectricity in intrafibrillar mineralization by comparing it to the case of WT collagen.

We qualitatively and quantitatively characterized the piezoelectric properties of collagen in WT and OI bone by utilizing advanced PFM techniques[31,43,44]. Particularly, we investigated the piezoelectric profiles along demineralized collagen fibrils extracted from both WT and OI bone. Then, they are compared with stiffness profiles of mineralized WT and OI collagen, which were previously reported[32,33,40–42]. Moreover, the piezoelectricity at the collagen matrix scale was examined as well as the morphology and structure. To the best of our knowledge, this is the first study investigating the piezoelectric property of OI collagen. This work also experimentally highlights the physiological importance of collagen piezoelectricity as a driving force responsible for ACP infiltration during the intrafibrillar mineralization process.

## Results and discussion

Figure 2 shows the qualitative PFM results measured on a demineralized collagen matrix extracted from WT and Amish OI mice bone (see Method section for details about sample preparation). First, resonance-enhanced PFM was employed to simultaneously characterize the topographic and piezoelectric features on the WT collagen matrix over an area of $0.5 \times 0.5\ \mu m^2$, as shown in Fig. 2a, b. The collagen fibrillar structure with the signature D-periodic spacing with repeated gap and overlap regions was clearly observed in the height map. The simultaneously obtained PFM amplitude map, where the bright part indicates higher piezoelectric amplitudes compared to the dark area, also exhibited a clear periodic pattern along the fibril on

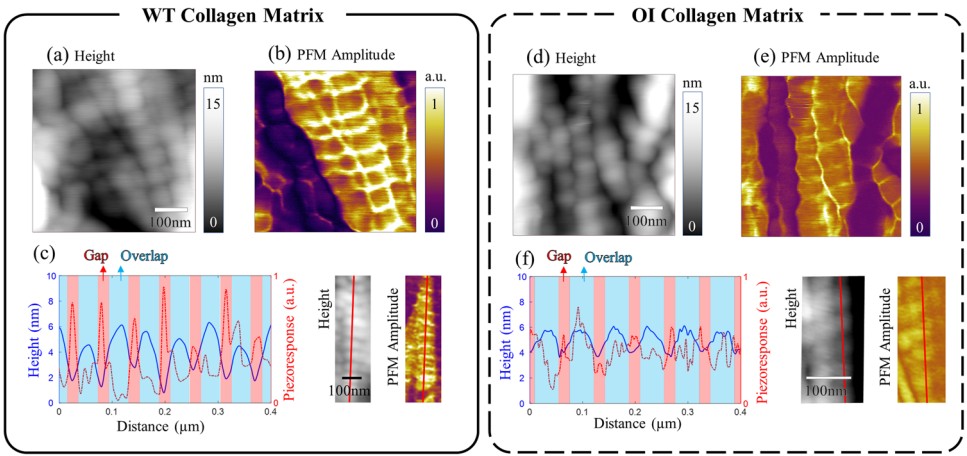

**Fig. 2 Structural and piezoelectric properties of WT and OI collagen measured by PFM. a** Height and **b** PFM amplitude maps of WT collagen matrix obtained by resonance-enhanced PFM, and **c** height and piezoresponse profiles of WT collagen fibril along the red lines shown in images on the right obtained by DART-PFM. **d** height and **e** PFM amplitude maps of OI collagen matrix obtained by resonance-enhanced PFM, and **f** height and piezoresponse profiles of OI collagen fibril along the red lines obtained by DART-PFM. The piezoelectric profile of the WT collagen revealed a clear periodic pattern along with the collagen fibrillar direction with peaks at each gap region, while the piezoelectric profile of the OI collagen was randomly varied along the fibril without strong correlation with the structure.

which the gap region showed higher PFM amplitudes than the overlap region. To better correlate fibril's piezoelectric pattern with its mechanical structure, we also employed a more advanced PFM technique, so-called dual AC resonance tracking PFM (DART-PFM)[44]. Conventional PFM suffers from crosstalk between topographic and piezoelectric information due to a shift of contact resonance frequency during scanning[45]. DART-PFM is designed to mitigate the crosstalk issue by compensating the resonance frequency shift. Even though DART-PFM sometimes cannot totally eliminate the crosstalk effect, the tracking technique minimizes the crosstalk and ambiguity between these two different types of information obtained by PFM. More discussion about the DART-PFM method and data analysis is shown in the Method section and Supplementary Information. The piezoelectric profile, obtained by DART-PFM, is plotted in Fig. 2c along the crest of a collagen fibril marked by a red line in the images on the right. Consistent with our previous results measured on a single collagen fibril extracted from bovine Achilles tendon[31], the piezoelectric profile showed a periodic pattern that is inversely correlated with the topographic profile repeating every D-spacing. Namely, the gap region of the fibrillar structure is 2–4 nm lower than the overlap region, while the piezoresponse amplitudes in the gap region are around 2-fold higher than the values in the overlap region.

The same types of PFM measurements were performed on the demineralized OI collagen as shown in Fig. 2d–f. The OI fibrillar structure shown in the height map (Fig. 2d) revealed a similar D-periodic structure with repeated gap and overlap regions. In the PFM amplitude map (Fig. 2e), however, no evident periodic pattern along the fibril manifested, indicating reduced heterogeneity in piezoelectricity. This trend is more clearly seen in the topographic and piezoelectric profiles, shown in Fig. 2f, obtained by DART-PFM. While the OI fibril exhibited an organized periodic profile with similar D-spacing in its structure, its piezoelectric variation was somewhat random, and no strong correlation was observed between the topographic and piezoelectric patterns.

Even though these DART-PRM results show a clear difference in the piezoresponse profile of WT and OI collagen with reduced possibility of topographic artifact, the results need to be confirmed using a method that is not affected by topographic variations. Thus, we also performed the quantitative measurement of

piezoresponse amplitude at a number of fixed points on the gap and overlap regions of each collagen while the applied AC voltage was varied from 0 V to 5 V in 0.5 V steps. The detailed calibration protocol to obtain the quantitative piezoresponse amplitude is described in the Method section. For statistical analysis, five collagen fibrils in each of the WT and OI samples were examined ($n = 5$) and the point measurement was repeated ten times. Figure 3a, b show the piezoresponse with respect to the input AC voltage, measured on the gap (red) and overlap (blue) regions of WT and OI collagens, respectively. Each data point shows the average of piezoresponse amplitude with its error of standard deviation, averaged over 50 measurements from five different fibrils. (Each fibril's piezoresponse amplitudes are shown in Supplementary Fig. 1). The piezoelectric coefficients at the gap and overlap regions were obtained from the slope of regression lines. In the WT collagen, the piezoelectric coefficient at the gap region was evaluated to be 0.31 ± 0.02 pm/V, which was about 55% higher than the value of 0.20 ± 0.01 pm/V obtained at the overlap region. Likewise, the piezoelectric coefficient of the OI collagen on the gap region, 0.37 ± 0.03 pm/V, was evaluated to be 16 % higher than that of the overlap region, 0.32 ± 0.03 pm/V. When the difference between these values was statistically examined using the post-hoc Tukey's test (Fig. 3c), the WT collagen revealed a significant difference between the piezoresponses at the gap and overlap regions ($p < 0.001$), while there was no such difference between the two regions in the OI collagen ($p > 0.05$). This result quantitatively supports the qualitative observation shown in the PFM maps (cf. Figure 2c, f) in that the OI collagen loses its piezoelectric heterogeneity within the fibril as well as the strong correlation with the structural properties. The boost in piezoresponse at OI's overlap region can be attributed to G610C substitutions. In a previous study, it was suggested that glycine substitutions of OI collagen cause misfolding of its triple-helix structure[46–48]. Subsequently, the unstable helical structure of tropocollagen may lead to higher susceptibility to structural change in response to electrical input or vice versa. This structure eventually would cause an increase of piezoelectricity at the overlap region in OI collagen.

When interpreted with the stiffness pattern in mineralized WT and OI collagen fibrils shown in the previous studies[32,33,40–42], these PFM results in demineralized WT and OI collagens suggest physiological implications about the role of collagen

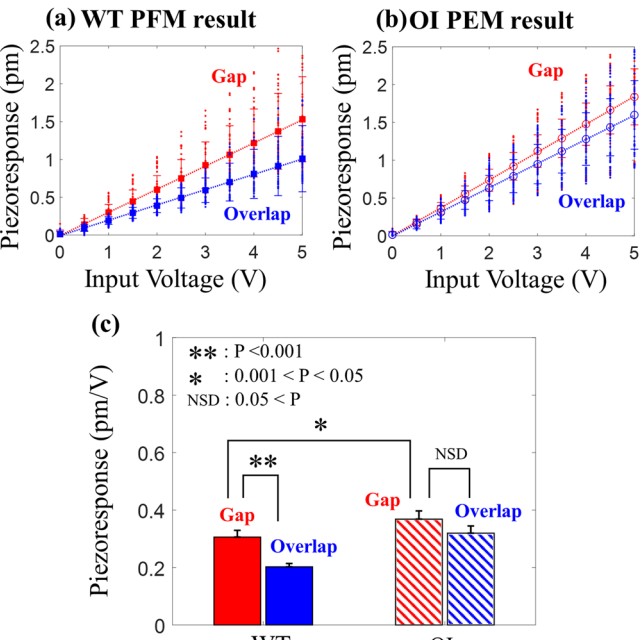

**Fig. 3 Comparison of piezoresponse measured at the gap and overlap regions of WT and OI collagens.** The PFM result of point measurements at the gap (red) and overlap (blue) regions of **a** WT and **b** OI collagen. The cantilever tip was precisely located at the gap and overlap regions and AC voltage was applied ranging from 0 V to 5 V. The red and blue lines indicate the average piezoresponse of five fibrils at the gap and overlap regions in both WT and OI collagen. **c** One-way ANOVA with a post-hoc Tukey HSD test was used to determine the statistically significant differences between the piezoresponse at the gap ($0.31 \pm 0.02$ pm/V) and overlap ($0.20 \pm 0.01$ pm/V) regions in WT collagen and at the gap ($0.37 \pm 0.03$ pm/V) and overlap ($0.32 \pm 0.03$ pm/V) regions in OI collagen. The significance was set at $0.001 < p\text{-value} < 0.05$ (*) and $p$-value $< 0.001$ (**).

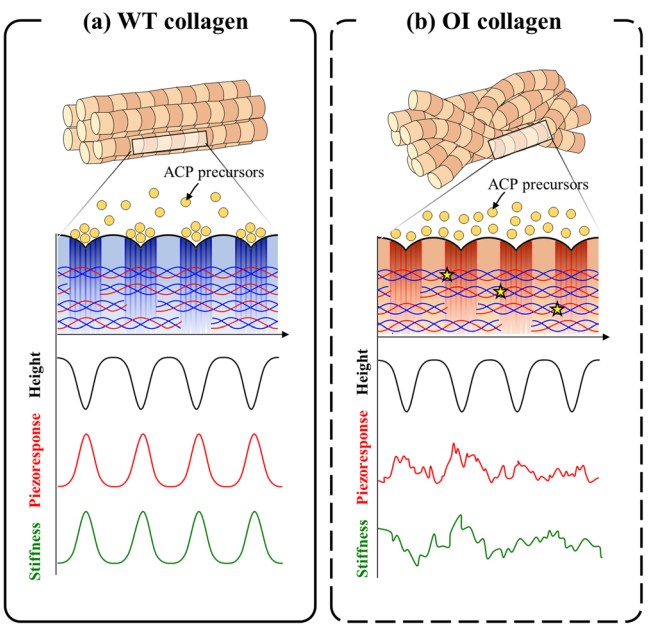

**Fig. 4 Schematic of the relationship between morphology, piezoresponse, and mechanical stiffness in collagen fibrils.** A schematic of relationship between morphology, piezoresponse, and mechanical stiffness of **a** WT collagen and **b** OI collagen along a fibril. It was reported that mineralized collagen in WT bone has higher stiffness at each gap zone, which corresponds with the piezoresponse pattern observed in this study. On the other hand, no such stiffness pattern of mineralized OI collagen was observed previously and, likewise, no piezoelectric pattern of OI collagen was measured, although the morphology of the OI collagen was similar to the WT collagen. This result emphasizes the role of collagen piezoelectricity for ACP infiltration through the gap region.

piezoelectricity in intrafibrillar mineralization. In one of the previous studies[33], dual-frequency AFM had been applied to characterize the mechanical stiffness of mineralized collagen fibrils from WT and OI mouse models. As illustrated in Fig. 4a, b, the WT collagen fibrils showed an ordered stiffness distribution inversely correlated with the fibrillar structure. This result had corroborated the fact that, in healthy bone, more mineral constituents are deposited at the gap region to locally stiffen this area where it was initially (i.e., in an unmineralized state) less stiff due to its lower molecular packing density[49–53]. This intrafibrillar mineralization is well known in that ACP mineral precursors infiltrate through the gap region and gradually crystalize along the fibril, even though there still exist debates about its driving force[26,54]. On the other hand, the OI collagen showed a random distribution of stiffness without an apparent relationship with the similarly ordered fibrillar structure[33]. This indicates that OI collagen suffers from abnormal intrafibrillar mineralization that causes the loss of nanomechanical heterogeneity along the collagen fibrils. Considering the fibrillar structure of OI collagen is not qualitatively dissimilar from the WT fibrillar structure, the less organized intrafibrillar mineralization occurring in OI bone indicates this process is not purely governed by collagen's structural or structure-related properties. Now when these stiffness profiles are compared with the piezoelectric profiles examined in our study, the periodic and random piezoelectricity profiles observed in the respective WT and OI collagen fibrils exactly match the mechanical stiffness patterns. This coincidence

of stiffness and piezoelectricity patterns implies the significance of piezoelectricity in intrafibrillar mineralization, especially as a mechanism explaining ACP infiltration into the gap zone. The phenomenon of ACP infiltration through gap regions has been explained by various driving forces, as mentioned in the introduction, such as electrostatic interaction, osmotic pressure, streaming potential, and NCPs[25–29]. These mechanisms, however, cannot sufficiently answer why the intrafibrillar mineralization in OI bone is not correlated with the periodic fibrillar structure of collagen with clear D-spacing. First, an amino acid substitution in the helical region of OI collagen would not affect its dipole moments in the telopeptide region, suggesting that electrostatic interactions between charged ACP and telopeptide region of collagen fibril should be comparable between the WT and OI collagen. Although loosely packed molecules can exist due to OI glycine substitutions, this is less likely to alter molecular concentrations between intra- and extrafibrillar space enough to induce a drastic change of the osmotic pressure profile across the fibril. Furthermore, because ionic strength and viscosity of bone fluid are not influenced by collagen mutation, streaming potential alone cannot explain why the intrafibrillar mineralization pattern is altered in OI bone. Lastly, there is no distinctive mechanism to alter the function of NCPs in OI collagen, especially considering that the overall fibrillar structure is qualitatively similar even with collagen mutation.

Collagen piezoelectricity has not garnered attention because the streaming potential better explains the mechanism of SGP in a physiological condition[22–24,55]. However, it has still been suggested that collagen piezoelectricity may collaborate with other

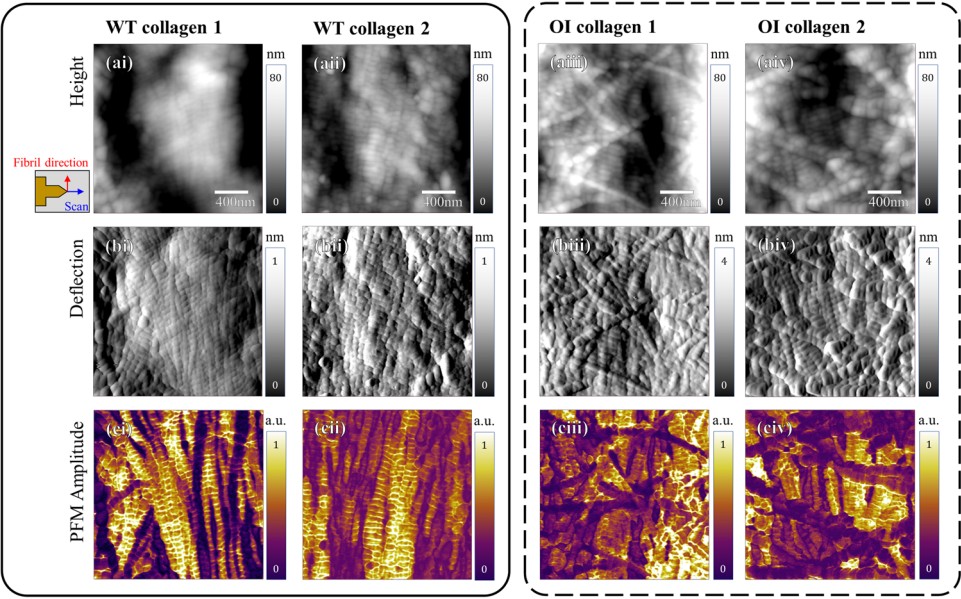

**Fig. 5 PFM measured on WT and OI collagen matrices.** PFM maps for **a** height, **b** deflection, and **c** PFM amplitude of WT and OI collagen matrices. In both WT and OI models, collagen fibrillar structures were exhibited with piezoelectricity. Although the WT collagen matrix was well-aligned in one direction, the OI collagen matrix exhibited a rather randomly aligned structure.

mechanisms discussed above[23]. Based upon the synchronous profiles between piezoelectricity and mechanical stiffness along the fibril, we suggest a perspective explaining the mechanism of intrafibrillar mineralization by emphasizing the role of collagen piezoelectricity. Here, collagen piezoelectricity would locally modulate the surface potential along the fibril with higher surface charges at the gap zone and, accordingly, provide a periodic pattern of zeta-potential on the collagen's surface. Because streaming potential is proportional to zeta-potential[55,56], a high zeta-potential induces an increase of hydraulic permeability to let ACP infiltrate through the gap region[23,57]. In the case of OI collagen, however, the non-periodic and relatively uniform distribution of piezoelectricity along a fibril would not provide local modulation of surface charges, resulting in less-ordered and less-heterogeneous intrafibrillar mineralization. Considering that bone heterogeneity is one of the main strategies to possess strength and ductility together for higher energy efficiency and better performance[5,58–60], losing heterogeneity at the fibrillar level would eventually result in brittleness and failure at high mechanical strains.

Figure 5 shows the structure and piezoelectricity of WT and OI collagen matrices measured over a larger fiber scale ($2 \times 2\ \mu m^2$) to observe the arrangement of collagen fibrils. From the height and deflection maps in Fig. 5a, b, the WT collagen matrix displays a well-aligned fibrillar structure in the vertical direction, while the OI collagen matrix exhibits a rather randomly aligned structure. Since collagen provides a template for mineral crystal growth along the fibril[61–63], the misalignment of OI fibrils can cause disoriented apatite formation. In the PFM amplitude maps shown in Fig. 5c, both WT and OI collagen matrices demonstrate local piezoresponse on the collagen fibrillar structure. The overall piezoelectric effect at the fiber level cannot be characterized using PFM, because the PFM technique only measures a local piezo-electric response underneath the AFM tip at the nanometer scale. Still, it can be inferred that piezoelectrically generated charges on each WT collagen fibril would be well correlated and accumulated between each other, thanks to collagen fibrils' ordered alignment, to provide a strong piezoelectric effect along the fibrillar direction

when longitudinal loading is applied to this structure. On the other hand, the overall piezoelectric effect of OI collagen fiber is possibly attenuated in OI bone due to the misalignment of OI collagen fibrils. Here, some collagen fibrils did not reveal a clear piezoresponse in PFM maps, especially when they were mis-aligned from the vertical direction (cf. dark-colored collagen fibrils in Fig. 5ciii, civ). This is because the PFM measurement depends on the direction of scanning due to the anisotropic nature of collagen piezoelectricity, causing a weak PFM amplitude of misaligned fibrils in the OI sample[64–67]. Some other low PFM amplitudes demonstrated by well-aligned fibrils are considered to be due to the surface condition, which may be less demineralized.

In addition to the alignment, another noticeable difference qualitatively observed in OI collagen matrices is a somewhat larger variation in the width of fibrils. To quantitatively confirm this, the width of 50 collagen fibrils in each model was measured, and their average and deviation from this measurement are plotted in Fig. 6a. As expected, the OI collagen shows a larger deviation of ± 31 nm from its average width of 109 nm, compared with the WT collagen which has a smaller deviation of ±19 nm from its average width of 107 nm. From this finding, we can infer that the G610C substitution in COL1A2 in OI model causes a problem in the cross-linking process which can be stratified from divalent to multivalent cross-links, building from an initial stag-gered assembly to a bigger 3D structure of collagen fibrils[68–70]. The observation that the OI collagen fibril exhibited a clear D-spacing with gap and overlap structures indicates that OI collagen fibril may not have a serious problem in its initial divalent cross-links. However, the varied width of OI fibrils implies a possible problem in the multivalent cross-linking process.

Finally, to investigate the effect of the extent of mineralization on collagen piezoelectricity, the mineralized and demineralized WT collagen samples were prepared and measured. As seen in Fig. 6b–e, the collagen matrix, originally covered by minerals, was revealed after eliminating minerals. The mineralized matrix did not exhibit strong piezoelectricity (MI, black, 0.08 ± 0.02 pm/V), while the demineralized collagen matrix displayed a clear

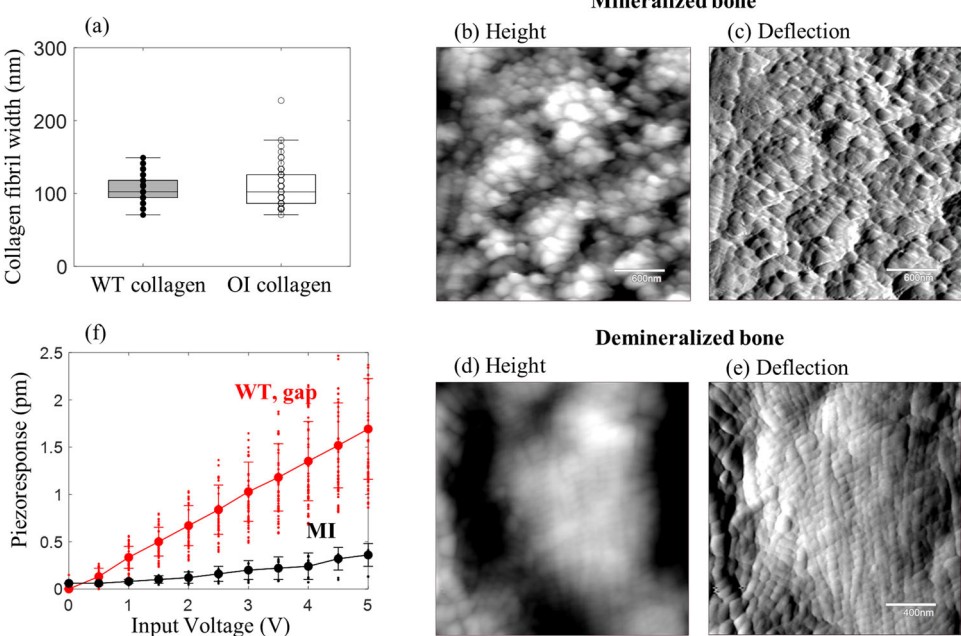

**Fig. 6 Comparison between mineralized and demineralized bone surfaces. a** Collagen fibril width of WT and OI models. In total, 50 fibril widths were investigated in each model. For WT collagen, 107 ± 19 nm width was observed, while 109 ± 31 nm width was observed for OI collagen. There was not a significant difference observed between WT and OI collagen width, as *p*-value was greater than 0.05 based on t-test analysis. **b** Height and **c** deflection maps of mineralized bone surface and **d** height and **e** deflection maps of the demineralized bone surface, respectively. **f** PFM results for mineralized (MI, black) and demineralized (WT, blue) bone surfaces. When compared to the collagen piezoresponse, the mineralized bone surface exhibited a very weak piezoelectric amplitude.

piezoresponse (WT, red, 0.31 ± 0.02 pm/V) as shown in Fig. 6f. Since the collagen fibrillar structure was not revealed, one point on the mineralized bone surface was randomly selected ($n = 1$), and the piezoresponse was measured as the same method of performing WT and OI collagen fibrils in Fig. 3. Each data point indicates the mean value of piezoresponse amplitude with its error of standard deviation. This result implies that the collagen piezoelectric effect can be weakened through mineralization. Thus, the collagen samples should be demineralized to investigate its piezoelectric response. Also, considering that the bone mineralization process is observed as two stages with a fast primary phase and a slow secondary phase[71–73], the collagen piezoelectricity would mainly affect the phase of primary mineralization. Additionally, no specific piezoelectric pattern was found on the fully mineralized collagen, which corresponds to the mechanical stiffness on fully mineralized tissues in a previous study[51]. After mineralized, the fibrillar structure disappears, revealing a uniformly distributed elastic modulus on the surface. Based on those findings, we can postulate the possible scenario that the collagen piezoelectricity would help ACP infiltration in the initial stage of intrafibrillar mineralization, but it may become less effective as collagen fibrils are mineralized.

## Conclusions
In summary, the role of collagen piezoelectricity in intrafibrillar mineralization as a driving force was profoundly evaluated by investigating the piezoresponse of WT and OI collagen samples. The PFM results from WT collagen displayed a periodic piezoelectric profile along collagen fibril's periodic structure with a larger value in gap regions (0.31 ± 0.02 pm/V) compared to the value in overlap regions (0.20 ± 0.01 pm/V). On the other hand, the OI collagen fibrils did not reveal a clear periodic pattern of piezoelectricity, exhibiting statistically similar values in the gap (0.37 ± 0.03 pm/V) and overlap regions (0.32 ± 0.03 pm/V),

despite the same structural pattern. Interestingly, these piezoelectric profiles on WT and OI collagen correspond to the profiles of mechanical stiffness in mineralized WT and OI collagen fibrils. This result implies that collagen piezoelectricity would locally modulate the surface potential along the fibril with higher surface charges at the gap zone, leading to an increase of zeta-potential and hydraulic permeability to help the ACP clusters infiltrate into the intrafibrillar space through the gap region. Therefore, the loss of mechanical heterogeneity in OI collagen can be attributed to the loss of piezoelectric heterogeneity, which finally results in a failure to high mechanical strains, ending up being more brittle. In this study, we also found that the WT collagen matrix displayed a fibrillar structure well-aligned in one direction, while the OI collagen matrix exhibited a rather randomly aligned structure. The misaligned collagen matrix in OI bone can also attenuate the piezoelectric effect when longitudinal loading is applied, which may also have an impact on the process of intra/inter-fibrillar mineralization. Because this collagen piezoelectricity becomes ineffective as the collagen matrix builds a large amount of mineral on its surface, it would play an important role at the intra- or inter-fibrillar level in the initial primary mineralization rather than the secondary mineralization. Although this study did not prove a direct relationship between the collagen piezoelectricity and mineralization, the physiological role of collagen piezoelectricity in intrafibrillar mineralization is plausibly supported to be considered as a perspective explaining the ACP infiltration.

## Method
**Animal model**. All samples used in this study were approved by IACUC regulations and by the laws and regulations of the USA. The Amish OI is a G610C knock-in mouse model, representing a human OI type IV phenotype, which alters the gly-610 codon (GGT) to a cysteine (TGT) codon for COL1A2[36,37]. The detailed information of OI mouse model is described in the Supplementary Information. In this study, two nine-week-old female Amish OI mice and two WT control mice were sacrificed. Femur bones were extracted from each model and soft tissues were stripped before processing for testing.

**Sample preparation**. To obtain the diaphysis part of the femur for imaging, the proximal and distal parts of the femur were removed by a low-speed sectioning saw. Then, each was polished by sandpaper (MicroCut™ Discs [P2500], BUEHLER) and abrasives (MetaDi™ Monocrystalline Suspension, 1 μm diameter, BUEHLER). The polished femurs were sonicated in deionized water for 30 seconds to remove polishing residue and debris. After cleaning, the femur was attached to a steel plate with conductive epoxy adhesive to provide an electrical ground across the sample. The prepared midshaft bone samples have 3~4 μm thickness. Next, the femur samples were treated with Ethylenediaminetetraacetic acid (UltraPure™ 0.5 M EDTA, pH 8.0, invitrogen™) solution for fourteen days on a rotating plate (Incu-Sharker Mini, Benchmark Scientific) at room temperature to eliminate the mineral part (i.e., hydroxyapatite) and expose the collagen structure. Lastly, the femur samples were cleaned with a sonicating again for 30 seconds to remove any residue. We kept the samples in the buffer solution and made it dried for 30 min before the measurement. During the experiment, the lab was maintained at the relative humidity of around 40% and the temperature of 20~23 °C.

**Resonance-enhanced PFM**. PFM, an application of AFM, allows simultaneous imaging of topography and piezoelectric responses of a specimen at the nanometer scale[74]. Traditional PFM applies an AC voltage to a sample through a conductive cantilever tip. The cantilever deflection at the applied frequency is detected to measure the piezoelectric response, while the static deflection is detected to image the topographic variations. In this work, we measured the shear piezoelectric coefficient ($d_{15}$) of collagen fibrils laying down on a conductive substrate by employing lateral (in-plane) PFM as illustrated in Fig. 7a. More details of the collagen piezoelectric tensor are described in the Supplementary Information.

Because a collagen fibril has a weak $d_{15}$ piezoelectric coefficient around 1 pm/V[64–66,75], it is challenging to measure it using traditional lateral PFM because of the low signal-to-noise ratio. Thus, we employed resonance-enhanced PFM to amplify the weak piezoresponse of collagen and characterize it quantitatively with a careful calibration process. The resonance-enhanced PFM applies an AC voltage at a contact resonance frequency to amplify the weak piezoresponse of a sample (Supplementary Fig. 2). The PFM amplitude and phase information can be obtained by a harmonic oscillator model described by

$$A(\omega) = \frac{a_{piezo}}{\sqrt{\left(1-\left(\frac{\omega}{\omega_0}\right)^2\right)^2 + \left(\frac{\omega}{Q}\cdot\omega_0\right)^2}}$$
$$\tan\phi(\omega) = \frac{\omega/\omega_0}{Q\left(1-\left(\frac{\omega}{\omega_0}\right)^2\right)} \tag{1}$$

where $\omega$ and $\omega_0$ are the drive frequency and contact resonance frequency, respectively, $a_{piezo}$ is the piezoresponse amplitude, and $Q$ is the quality factor of the contact resonance. When the drive frequency is equal to the contract resonance frequency ($\omega = \omega_0$), the piezoresponse amplitude ($a_{piezo}$) is amplified by the $Q$ factor (i.e., $A(\omega) = Qa_{piezo}$, and $\phi(\omega_0) = 90°$). For quantitative PFM, the resonance amplification by the quality factor should be considered when the actual piezoresponse amplitude ($a_{piezo}$) is calculated (see Quantitative PFM calibration section below).

Especially for resonance-enhanced PFM, it is important to address the parasitic artifacts originating from the electrostatic interaction between the cantilever and sample because it is amplified together. The electrostatic contribution to the PFM signal is described by[76]

$$A_{electrostatic} = \left| k^{-1}\left(\frac{dC}{dZ}\right)V_{ac}(V_{dc}-V_{sp}) \right| \cdot Q \tag{2}$$

where $k$ is the spring constant of a cantilever, $dC/dz$ is the capacitance derivative term in the direction between the tip and sample, $V_{ac}$ is the applied AC voltage, $V_{dc}$ is the applied DC voltage, and $V_{sp}$ is the surface potential. In our previous work[31], we measured $A_{electrostatic}$ while varying the DC voltage applied to the sample and found out $A_{electrostatic}$ is not varied by $V_{dc}$, demonstrating that the electrostatic artifact is negligible during lateral PFM. It is because the capacitance derivative term (i.e., $dC/dx$) is averaged out for the lateral movement of the cantilever.

**DART-PFM**. Resonance-enhanced PFM inherently suffers from the crosstalk effect between topographic and piezoelectric information because the contact resonance frequency shifts with variations of topography and material properties during scanning. To minimize such artifacts, lateral DART-PFM was employed in this work. As shown in Supplementary Fig. 3, DART-PFM applies two separate frequencies slightly higher and lower than the contact resonant frequency, and adjusts the drive frequencies to maintain measured amplitudes to be same by using a feedback loop[44]. A more detailed discussion about the method and measured data is shown in Supplementary Information and Supplementary Fig. 4.

**Quantitative PFM calibration**. In this study, the PFM was conducted with a commercial AFM system (MFP-3D Infinity, Asylum Research) using a conductive cantilever (2.5 N/m of nominal stiffness, 3XC-GG, OPUS®). For quantitative PFM with minimal artifacts, resonance-enhanced PFM was performed at the points on the gap and overlap regions of collagen fibrils, and the piezoresponse was quantified by following the careful calibration process.

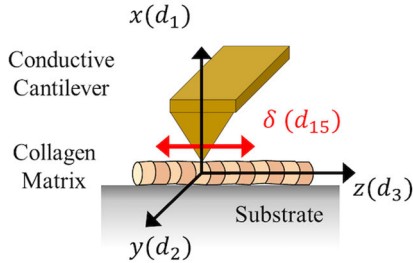

**(a) Lateral PFM**

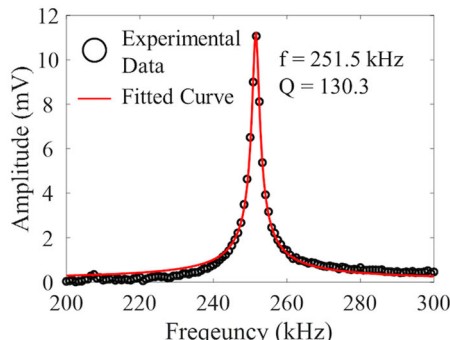

**(b) WT collagen resonant curve**

○ Experimental Data
— Fitted Curve
f = 251.5 kHz
Q = 130.3

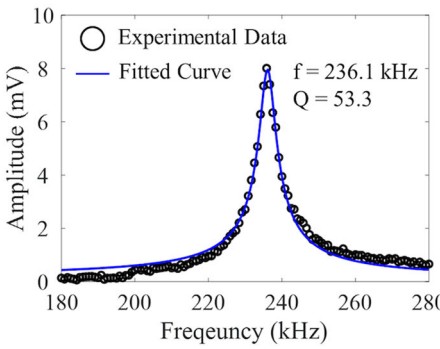

**(c) OI collagen resonant curve**

○ Experimental Data
— Fitted Curve
f = 236.1 kHz
Q = 53.3

**Fig. 7 Resonance-enhanced PFM. a** Sample coordinate for the lateral PFM. The cantilever was set in the y-axis to be perpendicular to the collagen fibrillar direction (z-axis). In this setting, the shear piezoelectric coefficient ($d_{15}$) of collagen can be obtained. The contact resonant curve of the first lateral mode measured on **b** WT collagen and **c** OI collagen. From the fitted curves, the contact resonance frequency and quality factor were obtained to be 251.5 kHz and 130.3 in the WT collagen and 236.1 kHz, and 53.3 in the OI collagen.

First, the cantilever was carefully calibrated in the lateral (in-plane) direction. We obtained the optical lever sensitivity (OLS) of the cantilever in the vertical direction from the force curves measured on a sapphire surface. The vertical OLS (V/nm) is a ratio of the sensing voltage of an AFM photodetector to the vertical deflection of the cantilever. Then, the lateral OLS can be estimated by the geometrical relationship between the vertical OLS and the lateral OLS, as described by[65–67,77]

$$R = \frac{\text{VinvOLS}}{\text{LinvOLS}} = \frac{\frac{\Delta z}{\Delta D_V}}{\frac{\Delta d}{\Delta D_L}} = \frac{\frac{L}{3S}}{\frac{h}{2S}} = \frac{2}{3}\cdot\frac{L}{h} \tag{3}$$

where $\Delta z$ and $\Delta d$ are out-of-plane and in-plane cantilever movements; $\Delta D_V$ and $\Delta D_L$ are the vertical and lateral movements of the laser spot on the photodiode; $L$ is the cantilever length, $S$ is the distance between the cantilever and the photodiode, and $h$ is the height of the tip. $R$ shows the ratio between the inverse OLS in the vertical (VinvOLS, nm/V) and lateral (LinvOLS, nm/V) directions for a convenient calculation. In this study, the same cantilever was employed to measure both WT and OI bone samples, and the VinvOLS was 172.94 nm/V, and R was 23.81 based on the probe geometry of $L = 500$ μm and $h = 14$ μm. Thus, the LinvOLS was determined to be 7.31 nm/V.

It is noteworthy that this OLS estimation was based on the static sensitivity of the cantilever, which is not always equal to the dynamic sensitivity when the cantilever is driven at or near its resonance[78]. It is because the dynamic sensitivity largely depends on the vibrational mode shape of the resonant cantilever which is also varied by the contact stiffness of the tip-surface junction ($k^*$). Following the method introduced in ref. 78, we estimated the ratio between the static and dynamic sensitivity of the cantilever by solving the cantilever vibration equation. The ratio between the dynamic OLS and static OLS ($\lambda$) is determined by

$$\lambda = \frac{\text{OLS}_{\text{dynamic}}}{\text{OLS}_{\text{static}}} = \frac{A_{\theta,c}(k^*)/Q_c(k^*)}{\theta_{static}} \quad (4)$$

where $A_{\theta,c}(k^*)$ is the amplitude of the cantilever slope at the tip position, $Q_c(k^*)$ is the Q factor at the contact resonance, and $\theta_{static}$ is the static slope corresponding to static cantilever displacement. $A_{\theta,c}(k^*)$ was numerically solved from the beam vibration equation with the contact stiffness $k^*$ attached to the tip. $k^*$ and $Q_c(k^*)$ are obtained experimentally. Consequently, we obtained the ratio $\lambda = 1.003$ from the experiment data on a collagen fibril with the contact resonance frequency ($f_c = 67.49 kHz$), Q factor (58.07), and the contact stiffness ($k^* = 18.63 N/m$) with 3XC-GG, OPUS® cantilever. Based on the result, we concluded that the difference in the static and dynamic sensitivities is negligible under the experimental condition.

Finally, the actual piezoelectric strain ($a_{piezo}$) can be quantitively determined by

$$a_{piezo} = \frac{A_{mv}}{Q} \cdot \text{LinvOLS} = \frac{A_{mv}}{Q} \cdot \frac{\text{VinvOLS} \cdot 3h}{2L} \quad (5)$$

where $A_{mv}$ is the measured PFM amplitude in millivolt at the contact resonance frequency. For each PFM measurement, the contact resonance curve was obtained and fitted by Eq. 1 to find the contact resonance frequency and Q factor. Figure 7b, c show a representative result of the contact resonance and fitted curve of the first lateral mode measured on WT and OI collagen. From the fitted curves, the contact resonance frequency ($\omega_0/2\pi$) and quality factor ($Q$) were obtained to be 251.5 kHz and 130.3 in the WT collagen and 236.1 kHz and 53.3 in the OI collagen.

To further validate our calibration method, we also conducted vertical PFM on a PPLN (Periodically Poled Lithium Niobate) sample and quantified its piezoelectric coefficient by following the same calibration method. When the obtained value was compared with the nominal value provided by a supplier, it showed good agreement (see Supplementary Methods and Supplementary Fig. 5). Also, for the statistical validation, error propagation and statistical analysis were performed. The details of the error propagation are shown in the Supplementary Information.

**Statistics and reproducibility**. JMP statistical software (SAS Institute) was used for statistical analysis of obtained data for PFM amplitudes at the gap and overlap in WT and OI collagen. Five collagen fibrils in each WT and OI bone were examined ($n = 5$), and points on each fibril were selected in the gap and overlap regions, respectively. In total, 110 PFM data points were obtained at each location, using inputs ranging from 0 V to 5 V and stepping by 0.5 V. At each input step, PFM result was obtained ten times for statistical analysis. Subsequently, the PFM results at the gap and overlap points in each WT and OI model were analyzed by one-way ANOVA with post-hoc Tukey's Honest Significant Difference (HSD) test. Statistical significance was set at $p < 0.05$ and a regression line was fitted to each data set to determine the piezoresponse coefficients and their standard deviations.

**Reporting summary**. Further information on research design is available in the Nature Portfolio Reporting Summary linked to this article.

## Data availability
The authors declare that all data generated or analyzed and codes during this study are available within the article and its Supplementary Data files. The files include raw data of WT and OI PFM results and MATLAB codes to plot the figures. A detailed description of each file is in the README file.

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

## Acknowledgements

This work was financially supported in part by National Science Foundation (PFI-TT-1827545) and OSU Material Research Seed Grant Program with funding from NSF-DMR-1420451. This financial support is gratefully acknowledged. We thank Dr. Kim (College of Dentistry, The Ohio State University) for his help with WT and OI bone sample preparation.

## Author contributions

H.C. conceived and supervised this project. J.K. prepared samples, conducted experiments and performed statistical analysis. H.C. and J.K. contributed to interpreting the data and co-writing of the manuscript.

## Competing interests

The authors declare no competing interests.
