## [Peer Review File · Communications Biology]

Reviewers' comments:

Reviewer #1 (Remarks to the Author):

Review of "The role of collagen piezoelectricity in regulating nanomechanical heterogeneity through intrafibrillar mineralization"

Ryan Wagner

Review of work: The authors compare PFM measurements on collagen affected by brittle bone disease to healthy wild-type collagen. The authors observe that the diseased collagen has a much weaker PFM amplitude signal than the healthy collagen. This is an interesting result that elucidates important aspects of this disease on collagen behavior. However, aspects of the interpretation, analysis, and quantification of the PFM data are flawed and should be corrected. Specific comments are enumerated below.

Comment # 1: The description of resonance-enhanced AFM does not state which vibrational mode of the cantilever is utilized (i.e. first flexural mode, first lateral mode, etc.). Without this information, the correctness of parts of the subsequent methodology cannot be evaluated. For example, if the authors are using a flexural resonance of the AFM cantilever to infer in-plane (lateral) motion of the sample surface this would be a novel methodology whose details would need to be explained and justified. Please state explicitly which vibrational mode of the cantilever is used in the resonance-enhanced AFM experiments.

Comment # 2: The optical lever sensitivity obtained from a static force curve does not directly apply for calculating amplitudes in resonance-enhanced PFM results (lines 68 – 73 in supplement). See Balke, Nina, Stephen Jesse, Pu Yu, Ben Carmichael, Sergei V. Kalinin, and Alexander Tselev. "Quantification of Surface Displacements and Electromechanical Phenomena via Dynamic Atomic Force Microscopy." *Nanotechnology* 27, no. 42 (September 15, 2016): 425707. <https://doi.org/10.1088/0957-4484/27/42/425707>.) Please review the provided reference and ensure the analysis of the vertical invOLS as applied to the resonance-enhanced PFM data is correct.

Comment # 3: A calculation based on the geometric parameters of the cantilever is used to convert vertical invOLS into lateral invOLS (Eqn S2). Such methodologies typically assume that the laser spot reflecting off the cantilever onto the photodiode is circular. However, the laser spot on the Asylum Research MFP3d AFM systems is an ellipse. Additionally, it is not clear to me that such equations can be applied to interpret resonance-enhanced PFM measurements. Please check the utilized equation and either correct it or more fully justify its use.

Comment # 4: I agree with the author's statement that the use of DART can reduce topography and mechanical coupling artifacts in resonance enhanced PFM measurements. However, DART does not eliminate the presence of such artifacts. For example, when characterizing two samples of different mechanical properties the resonance frequency will both shift to the left and the amplitude will reduce (or increase depending on the type of excitation) on the softer sample. I recommend including more discussion and analysis of the DART data to reduce the possibility of such artifacts. For example: "How much does the resonance frequency shift during the DART measurements?" and "How significant of a change in sample stiffness does this frequency shift correspond with?" If these changes are small, it is more likely that the author's current interpretation of the PFM amplitude data is correct.

Reviewer #2 (Remarks to the Author):

The manuscript is aimed at investigation of local piezoelectricity of collagen I and its role in controlling the mechanical properties of bones via intrafibrillar mineralization. The topic is quite

interesting, but I have serious concerns about the technical aspects of the used method (PFM), which makes me skeptical about interpretation of the obtained results. This inevitably raises concerns about the manuscript values on the conceptual level as well.

An increased PFM signal in the gap regions of the collagen fibrils in Fig 2b,c is interpreted by the authors by increased piezoelectricity, which is correlated with higher stiffness of this particular region, which in turn is attributed to the higher degrees of mineralization. I question the very first link of this logical chain. The authors wrote that the PFM DART mode could minimize the crosstalk between topography and PFM, hence supporting the authenticity of the PFM signal. This is not true. Imaging at resonance could actually amplify the topographic contribution to the PFM signal so that the topographic features could appear in the PFM image. It does not matter that the increased PFM is inversely correlated with topography - it could be still related strictly to the topographic variations. For this reason, the DART mode works better on the smooth surfaces. Furthermore, PFM is prone to a variety of artifacts due to the electrostatic contribution that could affect the probing tip displacement. This contribution could be particularly significant for samples with weak piezoelectricity, such as collagen, and could be affected by the dielectric constant of the sample, its thickness, type of tip coating, etc. In the particular case of data in Fig 2, variations of the PFM signal seem to be exactly due to the electrostatic artifacts. Stiffness variations deduced from resonant frequency variations in Ref 33, have to be verified by additional testing. Did the authors try measurements of the force-distance curves? Necessity for such verification stems from the fact that the resonance frequency could be affected by the surface topography (tip-sample contact area will be different in the gap and overlap regions). Because of these uncertainties, interpretation of the results is under question. This is not mentioning some minor omissions, such as no information on whether PFM is measuring in-plane or out-of-plane signals, poor citation of previous works of PFM of collagen, whether samples had the same thickness.

Reviewer #3 (Remarks to the Author):

The manuscript presents a piezoresponse AFM study to measure at high-spatial resolution the piezoelectricity of collagen fibers. Specifically, the manuscript compares the piezoelectrical response between gap and overlap regions of wild-type (WT) collagen fibers from collagen fibers obtained from osteogenesis imperfecta (OI) bones. The measurements reveal little differences between the piezoelectrical response of gap and overlap regions in OI collagen. That results is in contrast with the behaviour observed on WT collagen. The authors interpret the results as a consequence of the mineralization of collagen by calcium phosphate. More specifically, the manuscript claims a physiological role for collagen piezoelectricity. This is an attractive, original and relevant contribution. The novelty of the results makes this manuscript suitable for publication in *Communications Biology* after some revision.

1 It is not very clear, but the experiments were performed in air. It is well-established that the nanomechanical properties of collagen fibers do depend on the environment (buffer solution, air...). The manuscript should acknowledge it.

2 At its best, the manuscript shows a correlation between piezoelectricity and mineralization. However, this correlation is also observed in the stiffness measurements (Fig. 4). In my view, the available experimental data does not support a physiological relevance for the piezoelectrical properties of collagen.

3 The units were missing in Fig. 4.

4 To set the current scientific context on collagen characterization at the nanoscale, the authors might consider to cite a recent contribution on the nanomechanical properties of collagen nanoribbons: Gisbert, V. G. et al. *ACS Nano* 15, 1850-1857 (2021).

We would like to sincerely thank the reviewers for their constructive comments, which were extremely helpful to improve the paper. Our responses to reviewers' comments are given by the following notation:

- *reprinted comments from the reviewer written in italics*
- *our response written in blue*
- *revision in the paper highlighted by yellow.*

Reviewers' comments:

Reviewer #1 (Remarks to the Author):

Review of “The role of collagen piezoelectricity in regulating nanomechanical heterogeneity through intrafibrillar mineralization”

Review of work: The authors compare PFM measurements on collagen affected by brittle bone disease to healthy wild-type collagen. The authors observe that the diseased collagen has a much weaker PFM amplitude signal than the healthy collagen. This is an interesting result that elucidates important aspects of this disease on collagen behavior. However, aspects of the interpretation, analysis, and quantification of the PFM data are flawed and should be corrected. Specific comments are enumerated below.

→ After receiving the review, we learned that we did not include enough details about PFM experiment and failed to gain credibility of data and analysis. To address the reviewer's concerns and comments, we added the details about the experimental procedure and data analysis. Specialized in linear/nonlinear dynamics of micromechanical cantilevers, the authors have a good understanding about various factors that can lead significant artifacts and/or misinterpretation of PFM measurements. For credible and reliable PFM measurements and analysis on collagen fibrils, authors had developed an experimental protocol for collagen sample preparations for PFM, tip calibration for resonance-enhanced lateral PFM, removal of electrostatic effects, and comparison between quantitative point measurement and qualitative DART PFM, which had been published in ACS Biomaterials Science & Engineering (<https://doi.org/10.1021/acsbiomaterials.0c01314>). All the experiment shown in the current paper strictly follows the protocol developed in our previous work and we are highly confident that our measurement and data analysis are not flawed, but we lacked in providing the details. We fully addressed this issue in the revised version of the manuscript by adding the details about the experiment and analysis.

Comment #1: *The description of resonance-enhanced AFM does not state which vibrational mode of the cantilever is utilized (i.e. first flexural mode, first lateral mode, etc.). Without this information, the correctness of parts of the subsequent methodology cannot be evaluated. For example, if the authors are using a flexural resonance of the AFM cantilever to infer in-plane (lateral) motion of the sample surface this would be a novel methodology whose details would need to be explained and justified. Please state explicitly which vibrational mode of the cantilever is used in the resonance-enhanced AFM*

experiments.

Reply to comment #1. We thank you for this comment. In this work, all PFM experiments were conducted with the first lateral mode (in-plane). Since collagen has a piezoelectric tensor, as described below, because of the hexagonal/tetragonal symmetry of a collagen fibril, the shear piezoelectric coefficient (d_{15}) can be measured in our measurement setting as illustrated below in Figure R1.

$$d_{ij} = \begin{bmatrix} 0 & 0 & 0 & d_{14} & d_{15} & 0 \\ 0 & 0 & 0 & d_{15} & -d_{14} & 0 \\ d_{31} & d_{31} & d_{33} & 0 & 0 & 0 \end{bmatrix}$$

Collagen piezoelectric tensor

Figure R1: Sample coordinate for the lateral PFM (Fig. 7a)

In this measurement set, the cantilever was aligned with the y-axis to be perpendicular to the collagen fibrillar direction (z-axis), and the shear piezoelectric coefficient (d_{15}) of collagen was obtained. Other piezoelectric tensors (d_{31} , d_{33} , and d_{14}) cannot be obtained through this configuration because the electric potential cannot be applied along the fibril's longitudinal direction to measure d_{31} and d_{33} and a collagen fibril was fixed on the substrate resulting in nullification in the d_{14} direction.

We added this information in our main text and Figure 7 and the supplementary information as shown below.

[In main text p.12]

In this work, we measured the shear piezoelectric coefficient (d_{15}) of collagen fibrils laying down on a conductive substrate by employing lateral (in-plane) PFM as illustrated in Fig. 7a. More details of the collagen piezoelectric tensor are described in the supplementary information.

[In Supplementary Information]

Piezoelectric tensor for a collagen fibril

The piezoelectric tensor of a collagen fibril is described by Eqn. S6.

$$d_{ij} = \begin{bmatrix} 0 & 0 & 0 & d_{14} & d_{15} & 0 \\ 0 & 0 & 0 & d_{15} & -d_{14} & 0 \\ d_{31} & d_{31} & d_{33} & 0 & 0 & 0 \end{bmatrix} \quad \text{Eqn. S6}$$

In this work, The AFM cantilever was aligned with the y-axis to be perpendicular to the collagen fibrillar direction (z-axis). In this measurement setting, only d_{15} piezoelectric coefficient can be assessed by the lateral PFM (in-plane), while other piezoelectric coefficients (d_{31} , d_{33} , and d_{14}) cannot be obtained because the electric potential cannot be applied along the fibril's longitudinal direction to measure d_{31} and d_{33} and a collagen fibril was fixed on the substrate resulting in nullification in the d_{14} direction.

Comment #2: *The optical lever sensitivity obtained from a static force curve does not directly apply for calculating amplitudes in resonance-enhanced PFM results (lines 68 – 73 in supplement). See Balke, Nina, Stephen Jesse, Pu Yu, Ben Carmichael, Sergei V. Kalinin, and Alexander Tselev. "Quantification of Surface Displacements and Electromechanical Phenomena via Dynamic Atomic Force Microscopy." Nanotechnology 27, no. 42 (September 15, 2016): 425707. <https://doi.org/10.1088/0957-4484/27/42/425707>.) Please review the provided reference and ensure the analysis of the vertical invOLS as applied to the resonance-enhanced PFM data is correct.*

Reply to comment #2. Thanks for giving the comment and introducing the reference. We totally agree that the static optical lever sensitivity (OLS) may not be equal to the dynamic OLS. By using the method in the suggested paper, we estimated the shape factor $\lambda = s'_{dynamic}/s'_{static} = 1.003$, where $s'_{dynamic}$ and s'_{static} are the dynamic and static sensitivity, respectively, from the experiment data with the contact resonance frequency ($f_c = 67.49$ kHz), Q factor (58.07), and the contact stiffness ($k^* = 18.63$ N/m). Based on this estimated value, we concluded that static (s'_{static}) and dynamic (s') sensitivities of the cantilever we employed would be similar.

To further validate our calibration method, we also conducted quantitative PFM on a PPLN (Periodically Poled Lithium Niobate) sample using the same type of cantilever (3XC-GG, OPUS® cantilever) and compared the obtained piezoelectric coefficient with the value provided by the supplier. For PPLN, we obtained the shape factor $\lambda = s'_{dynamic}/s'_{static} = 1.0355$ from the experiment data with the contact resonance frequency ($f_c = 76.87$ kHz), Q factor (67.6), and the contact stiffness ($k^* = 24.26$ N/m). Following the calibration protocol, we obtained 12 pm/V of d_{33} piezoelectric coefficient of which value is in the range of specification (PPLN d_{33} : 10~20 pm/V) provided by Asylum Research (<https://afm.oxinst.com/assets/uploads/products/asylum/documents/Piezoresponse-Force-Microscopy-AFM-web.pdf>).

We added the detailed information of the experiment validating our calibration method in our supplementary information as shown below.

[In the main text p.15]

It is noteworthy that this OLS estimation was based on the static sensitivity of the cantilever, which is not always equal to the dynamic sensitivity when the cantilever is driven at or near its resonance⁷⁷. It is because the dynamic sensitivity largely depends on the vibrational mode shape of the resonant cantilever which is also varied by the contact stiffness of the tip-surface junction (k^*). Following the method introduced in ref. 77, we estimated the ratio between the static and dynamic sensitivity of the cantilever by solving the cantilever vibration equation. The ratio between the dynamic OLS and static OLS (λ) is determined by

$$\lambda = \frac{\text{OLS}_{\text{dynamic}}}{\text{OLS}_{\text{static}}} = \frac{A_{\theta,c}(k^*)/Q_c(k^*)}{\theta_{\text{static}}} \quad \text{Eqn. 4}$$

where $A_{\theta,c}(k^*)$ is the amplitude of the cantilever slope at the tip position, $Q_c(k^*)$ is the Q factor at the contact resonance, and θ_{static} is the static slope corresponding to static cantilever displacement. $A_{\theta,c}(k^*)$ was numerically solved from the beam vibration equation with the contact stiffness k^* attached to the tip. k^* and $Q_c(k^*)$ are obtained experimentally. Consequently, we obtained the ratio $\lambda = 1.003$ on a collagen fibril from the experiment data with the contact resonance frequency ($f_c = 67.49 \text{ kHz}$), Q factor (58.07), and the contact stiffness ($k^* = 18.63 \text{ N/m}$) with 3XC-GG, OPUS[®] cantilever. Based on the result, we concluded that the difference in the static and dynamic sensitivities is negligible under the experimental condition.

[In Supplementary Information]

PFM experiment result of PPLN sample for validation of the calibration method

(a) PFM Amplitude
With Tip DC 120 mV

(b) PFM amplitude profile

Supplementary Fig. S4. (a) PFM amplitude map of a PPLN sample scanned by vertical resonance-enhanced PFM. (out-of-plane) (b) PFM amplitude profile along with the red line in (a). In these results, V_{invOLS} (185.6 nm/V) was applied to convert the photodetector voltage to the cantilever tip movement.

In order to validate our calibration method, we conducted the resonance-enhanced PFM on a PPLN (Periodically Poled Lithium Niobate) sample that has typically 10~20 pm/V (d_{33}) piezoelectric coefficient as provided by Asylum Research¹³. Here, we used the same type of cantilever that used

for the collagen sample (3XC-GG, OPUS®). The static sensitivity of the cantilever VinvOLS was obtained as 185.6 nm/V from the force curve on a sapphire surface. In addition, the cantilever stiffness and Q factor in free vibration were 2.1 N/m and 70.1, respectively, which were obtained from the thermal spectrum of the cantilever. On the PPLN surface, we obtained contact resonant frequency, Q factor, and contact stiffness as 76.87 kHz, 67.6, and 24.25 N/m, respectively. By numerically solving the beam vibration equation¹⁴, we obtained the ratio between the dynamic and static sensitivities of the cantilever as shown in Eqn.4 as $\lambda = 0.96$. Therefore, we assumed that the dynamic sensitivity equals the static sensitivity of the cantilever in this PFM setting. Because vertical PFM is greatly affected by the electrostatic force, 120 mV DC voltage (V_{dc}) in addition to 1 V AC input voltage (V_{ac}) was applied to compensate the surface potential (V_{sp}) to eliminate the electrostatic contribution based on Eqn. 2. Supplementary Fig. 5a displays the PFM amplitude map of the PPLN, and Supplementary Fig. 5b demonstrates the PFM amplitude profile along with the redline in Supplementary Fig. 5a. By calibrating the measured PFM amplitude, obtained piezoelectricity in d_{33} of the PPLN was obtained to be around 12 pm/V, which is in the range of the specification, supporting the validation of our calibration method.

Comment # 3: A calculation based on the geometric parameters of the cantilever is used to convert vertical invOLS into lateral invOLS (Eqn S2). Such methodologies typically assume that the laser spot reflecting off the cantilever onto the photodiode is circular. However, the laser spot on the Asylum Research MFP3d AFM systems is an ellipse. Additionally, it is not clear to me that such equations can be applied to interpret resonance-enhanced PFM measurements. Please check the utilized equation and either correct it or more fully justify its use.

Reply to comment 3. Thanks for pointing it out. This comment thankfully provided us a chance to review our calibration method closely. Because this method is based on a change of the laser spot as shown in Figure R2, we believe that the shape of the laser spot does not affect the sensitivity unless the laser shape is asymmetric [ref 65-67, 78].

Figure R2: Schematic of optical lever sensitivity in vertical (out-of-plane)

Here, the OLS in vertical is determined by $V_{invOLS} = \frac{\Delta z}{\Delta D}$. Since the angle of the cantilever at the tip can be derived by a beam equation $\theta = \frac{FL^3}{2EI}$, where F is the applied force to the cantilever tip, L is the length of the cantilever, E is Young's modulus of the cantilever, and I is the moment of inertia. Because $\Delta z = \frac{FL^3}{3EI}$, the angle of the cantilever is $\theta = \frac{3\Delta z}{2L}$.

Therefore, ΔD can be determined by $\Delta D = \sin(2\theta) \cdot S \approx 2\theta \cdot S = 3S \cdot \frac{\Delta z}{L}$. Where S means the displacement of the laser spot from the cantilever to the photodiode. As a result, we can get $V_{invOLS} = \frac{\Delta z}{\Delta D} = \frac{L}{3S}$. In this calculation, not the shape of the laser spot but the change of the laser spot was considered.

The lateral movement of the cantilever (In-plane) is shown in Figure R3.

Figure R3: Schematic of optical lever sensitivity in lateral (In-plane)

Similar to the previous case, the change of laser spot on the photodiode because of the lateral movement of the cantilever can be defined by $\Delta D = \sin(2\alpha) \cdot S \approx 2\alpha \cdot S$. Also, because $\tan \frac{d}{h} \approx \alpha$, the inverse optical lever sensitivity in lateral is $L_{invOLS} = \frac{d}{\Delta D} = \frac{\alpha h}{2\alpha S} = \frac{h}{2S}$. Again, not the shape of the laser spot but the change of the laser spot was considered in this estimation.

Consequently, the relationship between the vertical and lateral optical lever sensitivity can be defined by the geometrical relationship of the cantilever as shown below, and R is not affected by the shape of the as far as it is symmetric about x- and y-axis.

$$R = \frac{V_{invOLS}}{L_{invOLS}} = \frac{\frac{\Delta z}{\Delta D_V}}{\frac{d}{\Delta D_L}} = \frac{L}{3S} \cdot \frac{2}{3} \cdot \frac{L}{h} \quad \text{Eqn. 3.}$$

We also added more details and references of this converting part in the main text as shown below.

[In main text p.14]

First, the cantilever was carefully calibrated in the lateral (in-plane) direction. We obtained the optical lever sensitivity (OLS) of the cantilever in the vertical direction from the force curves measured on a sapphire surface. The vertical OLS (V/nm) is a ratio of the sensing voltage of an AFM photodetector to the vertical deflection of the cantilever. Then, the lateral OLS can be estimated by the geometrical relationship between the vertical OLS and the lateral OLS as defined by^{65-67,78}.

$$R = \frac{V_{invOLS}}{L_{invOLS}} = \frac{\frac{\Delta z}{\Delta D_V}}{\frac{\Delta d}{\Delta D_L}} = \frac{L}{3S} = \frac{2}{3} \cdot \frac{L}{h} \quad \text{Eqn. 3}$$

where Δz and Δd are out-of-plane and in-plane cantilever movements; ΔD_V and ΔD_L are the vertical and lateral movements of the laser spot on the photodiode; L is the cantilever length, S is the distance between the cantilever and the photodiode, and h is the height of the tip. R shows the ratio between the inverse OLS in the vertical (V_{invOLS} , nm/V) and lateral (L_{invOLS} , nm/V) directions for a convenient calculation. In this study, the same cantilever was employed to measure both WT and OI bone samples, and the V_{invOLS} was 172.94 nm/V, and R was 23.81 based on the probe geometry of $L=500 \mu\text{m}$ and $h=14 \mu\text{m}$. Thus, the L_{invOLS} was determined to be 7.31 nm/V.

In addition, we added details of how we got the quantitative piezoresponse in the main text and Fig. 7b and c.

[In main text p.16]

Finally, the actual piezoelectric strain (a_{piezo}) can be quantitatively determined by

$$a_{piezo} = \frac{A_{mv}}{Q} \cdot L_{invOLS} = \frac{A_{mv}}{Q} \cdot \frac{V_{invOLS} \cdot 3h}{2L} \quad \text{Eqn. 5}$$

where A_{mv} is the measured PFM amplitude in millivolt at the contact resonance frequency. For each PFM measurement, the contact resonance curve was obtained and fitted by Eqn. 1 to find the contact resonance frequency and Q factor. Figures 7b and c show a representative result of the contact resonance and fitted curve of the first lateral mode measured on WT and OI collagen, respectively. From the fitted curves, the contact resonance frequency ($\omega_0/2\pi$) and quality factor (Q) were obtained to be 251.5 kHz and 130.3 in the WT collagen and 236.1 kHz and 53.3 in the OI collagen.

Figure 7(b),(c)

Comment # 4: I agree with the author's statement that the use of DART can reduce topography and mechanical coupling artifacts in resonance enhanced PFM measurements. However, DART does not eliminate the presence of such artifacts. For example, when characterizing two samples of different mechanical properties the resonance frequency will both shift to the left and the amplitude will reduce (or increase

depending on the type of excitation) on the softer sample. I recommend including more discussion and analysis of the DART data to reduce the possibility of such artifacts. For example: “How much does the resonance frequency shift during the DART measurements?” and “How significant of a change in sample stiffness does this frequency shift correspond with?” If these changes are small, it is more likely that the author’s current interpretation of the PFM amplitude data is correct.

Reply to comment 4. We totally agree with your comment. We added the shift of resonant frequency data in the supplementary information, as shown below.

Supplementary Figure 4 shows the height, piezoresponse, and resonant frequency profile of (a) WT collagen and (b) OI collagen from DART-PFM. Although, in both collagens, the resonant frequency was shifted, the DART-PFM was able to track and compensate for the resonant frequency shifting. Also, the resonant frequency shifting patterns do not match the height and piezoresponse profiles, indicating that the piezoresponse of each collagen does not stem from the resonant frequency change.

It is also very important to emphasize that we also quantitatively measured the piezoresponse amplitude at the gap and overlap regions by employing point measurements of PFM, rather than scanning the area. These point measurement results are not affected by the variations of height and material properties, because each data point is calibrated to eliminate such artifacts. As shown in the results, the DART-PFM profiles on OI and WT collagen are well supported by the point measurement data.

To address the reviewer’s comment, we added the limitation of the DART-PFM technique and the resonance frequency shifting data into the main text and supplementary information, as shown below.

[In the main text p. 4]

Even though DART-PFM cannot totally ignore the crosstalk effect, it can mitigate the crosstalk issue by compensating the resonance frequency shift. This tracking technique minimizes the crosstalk and ambiguity between these two different types of information obtained by PFM. More discussion about DART-PRM is shown in the method section and Supplementary Information.

[In the main text p. 5]

Even though these DART-PRM results show a clear difference in the piezoresponse profile of WT and OI collagen with reduced possibility of topographic artifact, the results need to be confirmed using a method that is not affected by topographic variations. Thus, we also performed the quantitative measurement of piezoresponse amplitude at a number of fixed points on the gap and overlap regions of each collagen while the applied AC voltage was varied from 0V to 5 V in 0.5V steps. The detailed calibration protocol to obtain the quantitative piezoresponse amplitude is described in the method section.

[In the Supplementary Information]

Supplementary Fig. 4 shows the height, piezoresponse, and resonant frequency profile along with the WT and OI collagen fibril. Although, in both collagens, the resonant frequency was shifted,

the DART-PFM was able to track and compensate for the resonant frequency shifting. Also, the resonant frequency shifting patterns do not match the height and piezoresponse profiles, indicating that the piezoresponse of each collagen does not stem from the resonant frequency change. Moreover, the piezoresponse patterns in the profile of both collagen samples acquired by the DART-PFM (Fig.2c and f) support the quantitative results obtained by the resonance-enhanced PFM (Fig. 3).

Supplementary Fig. 4. Height, piezoresponse, and resonant frequency profile of (a) WT collagen and (b) OI collagen obtained from DART-PFM.

Reviewer #2 (Remarks to the Author):

The manuscript is aimed at investigation of local piezoelectricity of collagen I and its role in controlling the mechanical properties of bones via intrafibrillar mineralization. The topic is quite interesting, but I have serious concerns about the technical aspects of the used method (PFM), which makes me skeptical about interpretation of the obtained results. This inevitably raises concerns about the manuscript values on the conceptual level as well.

1. An increased PFM signal in the gap regions of the collagen fibrils in Fig 2b,c is interpreted by the authors by increased piezoelectricity, which is correlated with higher stiffness of this particular region, which in turn is attributed to the higher degrees of mineralization. I question the very first link of this logical chain. The authors wrote that the PFM DART mode could minimize the crosstalk between topography and PFM, hence supporting the authenticity of the PFM signal. This is not true. Imaging at resonance could actually amplifies the topographic contribution to the PFM signal so that the topographic features could appear in the PFM image. It does not matter that the increased PFM is inversely correlated with topography - it could be still related strictly to the topographic variations. For this reason, the DART mode works better on the smooth surfaces.

Reply to comment 1. We appreciate this valuable comment. We agree with the reviewer in that DART-PFM may not completely eliminate the crosstalk, and it is more reliable on a smooth and homogeneous surface. However, we want to emphasize that the trend measured by DART-PFM (i.e., higher piezoresponse on gap vs. lower piezoresponse on overlap in WT collagen; rather homogenous piezoresponse in OI collage) is well supported by the quantitative point measurement data shown in Figure 3. These point measurements are not affected by the topography or mechanical properties of the sample, because each point is calibrated by the resonance information measured each time, as described in the method section.

Additionally, to validate the DART-PFM result, we added resonant frequency shifting data and discussion in the Supplementary Information. Supplementary Figure 4 shows the height, piezoresponse, and resonant frequency profile of (a) WT collagen and (b) OI collagen from DART-PFM. Although, in both collagens, the resonant frequency was shifted, the DART-PFM was able to track and compensate for the resonant frequency shifting. Also, the resonant frequency shifting patterns do not match the height and piezoresponse profiles, indicating that the piezoresponse of each collagen does not stem from the resonant frequency change.

To address the reviewer's comment, we added the limitation of the DART-PFM technique and the resonance frequency shifting data into the main text and supplementary information, as shown below.

[In the main text p. 4]

Even though DART-PFM cannot totally ignore the crosstalk effect, it can mitigate the crosstalk issue by compensating the resonance frequency shift. This tracking technique minimizes the

crosstalk and ambiguity between these two different types of information obtained by PFM. More discussion about DART-PRM is shown in the method section and Supplementary Information.

[In the main text p. 5]

Even though these DART-PRM results show a clear difference in the piezoresponse profile of WT and OI collagen with reduced possibility of topographic artifact, the results need to be confirmed using a method that is not affected by topographic variations. Thus, we also performed the quantitative measurement of piezoresponse amplitude at a number of fixed points on the gap and overlap regions of each collagen while the applied AC voltage was varied from 0V to 5 V in 0.5V steps. The detailed calibration protocol to obtain the quantitative piezoresponse amplitude is described in the method section.

[In the Supplementary Information]

Supplementary Fig. 4 shows the height, piezoresponse, and resonant frequency profile along with the WT and OI collagen fibril. Although, in both collagens, the resonant frequency was shifted, the DART-PFM was able to track and compensate for the resonant frequency shifting. Also, the resonant frequency shifting patterns do not match the height and piezoresponse profiles, indicating that the piezoresponse of each collagen does not stem from the resonant frequency change. Moreover, the piezoresponse patterns in the profile of both collagen samples acquired by the DART-PFM (Fig.2c and f) support the quantitative results obtained by the resonance-enhanced PFM (Fig. 3).

Supplementary Fig. 4. Height, piezoresponse, and resonant frequency profile of (a) WT collagen and (b) OI collagen obtained from DART-PFM.

2. Furthermore, PFM is prone to a variety of artifacts due to the electrostatic contribution that could affect the probing tip displacement. This contribution could be particularly significant for samples with weak piezoelectricity, such as collagen, and could be affected by the dielectric constant of the sample, its thickness, type of tip coating, etc. In the particular case of data in Fig 2, variations of the PFM signal seem to be exactly due to the electrostatic artifacts.

Reply to comment 2. We totally agree with this comment that the electrostatic contribution should be carefully addressed. In our previous work, we investigated the electrostatic contribution on PFM results, and we successfully demonstrated that the lateral PFM is not affected by the electrostatic force (Ref. 31). This is because the capacitance derivative term (dC/dZ) in the electrostatic equation is averaged out during the lateral PFM (Ref. 72).

$$A_{electrostatic} = \left| k^{-1} \left(\frac{dC}{dz} \right) V_{ac} (V_{dc} - V_{sp}) \right| \cdot Q$$

To examine and eliminate the electrostatic effect, we measure the PFM amplitude with varied DC voltage (V_{dc}) as shown in Figure R4 (brought from our previous paper, ref.31). In vertical (out-of-plane) PFM (cf. Fig. R4c left), a V-shape of PFM amplitude was revealed depending on V_{dc} because of the electrostatic interaction. In lateral (in-plane) PFM (cf. Fig. R4c right), however, the PFM amplitude was not varied by V_{dc} , indicating that lateral PFM is not affected by electrostatic interactions.

Figure R4: The effect of electrostatic forces on vertical PFM and lateral PFM

To address this comment, we added this information in the main text as shown below.

[In the main text p.13]

Especially for resonance-enhanced PFM, it is important to address the parasitic artifacts originating from the electrostatic interaction between the cantilever and sample because it is amplified together. The electrostatic contribution to the PFM signal is described by ⁷⁶

$$A_{electrostatic} = \left| k^{-1} \left(\frac{dC}{dz} \right) V_{ac} (V_{dc} - V_{sp}) \right| \cdot Q \quad \text{Eqn. 2}$$

where k is the spring constant of a cantilever, dC/dz is the capacitance derivative term in the direction between the tip and sample, V_{ac} is the applied AC voltage, V_{dc} is the applied DC voltage, and V_{sp} is the surface potential. In our previous work³¹, we measured $A_{electrostatic}$ while varying the DC voltage applied to the sample and found out $A_{electrostatic}$ is not varied by V_{dc} , demonstrating that the electrostatic artifact is negligible during lateral PFM. It is because the capacitance derivative term (i.e., dC/dx) is averaged out for the lateral movement of the cantilever.

3. Stiffness variations deduced from resonant frequency variations in Ref 33, have to be verified by additional testing. Did the authors try measurements of the force-distance curves? Necessity for such verification stems from the fact that the resonance frequency could be affected by the surface topography (tip-sample contact area will be different in the gap and overlap regions). Because of these uncertainties, interpretation of the results is under question.

Reply to comment 3. We appreciate this comment. We agree with the uncertainties of AMFM technique utilized in Ref 33. Unfortunately, however, performing stiffness verification in this work may not be relevant because the mineralization status of collagen samples between these two works is different. In this work, we chemically *demineralized* the WT and OI bone to eliminate the minerals and it is well known that the stiffness at the gap region is lower in the demineralized collagen. However, the samples used in Ref. 33 was not chemically demineralized, which maintains the stiffness variations of a mineralized collagen fibril (higher at the gap region).

It is well known that the stiffness at the gap region of WT collagen is higher, as observed in Ref. 33, since the mineral precursors are infiltrated into and accumulated at the gap regions during the intrafibrillar mineralization. Moreover, the loss of intrafibrillar mineralization in OI bone has been also reported by employing SAXS (Small-Angle X-ray Scattering), supporting the result in Ref. 33. There are also several papers revealing the stiffness variations in WT and OI collagen by employing other characterization techniques, supporting the result in Ref. 33.

Pick-force-tapping mode AFM

1. Liu, Y. et al. Hierarchically Staggered Nanostructure of Mineralized Collagen as a Bone-Grafting Scaffold. *Adv. Mater.* 28, 8740–8748 (2016).

This graph demonstrated Young's modulus of mineralized WT collagen by employing the pick-force tapping mode AFM. The periodic distribution of the modulus with ~ 67 nm is identified in (g) (upper), and the section analysis shows a higher modulus in the gap zone of collagen and a lower modulus in the overlap zone of collagen in (g) (lower).

Collagen intrafibrillar mineralization on TEM

1. Liu, Y. et al. Intrafibrillar Collagen Mineralization Produced by Biomimetic Hierarchical Nanoapatite Assembly. *Advanced Materials* 23, 975–980 (2011).
2. Niu, L. et al. Multiphase Intrafibrillar Mineralization of Collagen. *Angewandte Chemie International Edition* 52, 5762–5766 (2013).

SAXS (Small-angle X-ray scattering) in WT and OI

1. Maghsoudi-Ganjeh, M., Samuel, J., Ahsan, A. S., Wang, X. & Zeng, X. Intrafibrillar mineralization deficiency and osteogenesis imperfecta mouse bone fragility. *Journal of the Mechanical Behavior of Biomedical Materials* 117, 104377 (2021).
2. Fratzl, P., Paris, O., Klaushofer, K. & Landis, W. J. -Bone mineralization in an osteogenesis imperfecta mouse model studied by small-angle x-ray scattering. *J. Clin. Invest.* 97, 396–402 (1996).

To address the reviewer's comment, we added this information into the main text as shown below.

[In the main text p.3]

Many studies utilizing AFM (Atomic Force Microscopy), TEM (Transmission electron microscopy), and SAXS (Small-angle X-ray scattering) confirmed that OI bone undergoes abnormal mineralization, and the OI collagen loses its heterogeneity of mechanical stiffness along the fibrils from the fibrillar level^{32,33,40–42}.

[In the main text p.4]

Particularly, we investigated the piezoelectric profiles along demineralized collagen fibrils extracted from both WT and OI bone. Then, they are compared with stiffness profiles of mineralized WT and OI collagen, which were previously reported^{32,33,40–42}.

Also, we emphasized the abnormal intrafibrillar mineralization in OI collagen in the main text as seen below.

[In the main text p.6]

When interpreted with the stiffness pattern in mineralized WT and OI collagen fibrils shown in the previous studies^{32,33,40–42}, these PFM results in demineralized WT and OI collagens suggest significant physiological implications about the role of collagen piezoelectricity in intrafibrillar mineralization. In one of the previous studies³³,

[In the main text p.7]

On the other hand, the OI collagen showed a random distribution of stiffness without an apparent relationship with the similarly ordered fibrillar structure³³. This indicates that OI collagen suffers from abnormal intrafibrillar mineralization that causes the loss of mechanical heterogeneity along the collagen fibrils

4. This is not mentioning some minor omissions, such as no information on whether PFM is measuring in-plane or out-of-plane signals, poor citation of previous works of PFM of collagen, whether samples had the same thickness.

Reply to comment 4. We thank you for pointing this out. We added the details of experimental information in the main text and supplementary information as shown below.

For the citation of previous PFM works on collagen, we had to omit some references due to the limit in the number of references of this journal. To the best of our knowledge, this is the first study investigating the piezoelectricity in OI collagen, while few attempts have been conducted to investigate the WT collagen piezoelectricity that we included in our paper.

[In main text p.12]

In this work, we measured the shear piezoelectric coefficient (d_{15}) of collagen fibrils laying down on a conductive substrate by employing lateral (in-plane) PFM as illustrated in Fig. 7a. More details of the collagen piezoelectric tensor are described in the supplementary information.

[In Supplementary Information]

Piezoelectric tensor for a collagen fibril

The piezoelectric tensor of a collagen fibril is described by Eqn. S6.

$$d_{ij} = \begin{bmatrix} 0 & 0 & 0 & d_{14} & d_{15} & 0 \\ 0 & 0 & 0 & d_{15} & -d_{14} & 0 \\ d_{31} & d_{31} & d_{33} & 0 & 0 & 0 \end{bmatrix} \quad \text{Eqn. S6}$$

In this work, The AFM cantilever was aligned with the y-axis to be perpendicular to the collagen fibrillar direction (z-axis). In this measurement setting, only d_{15} piezoelectric coefficient can be assessed by the lateral PFM (in-plane), while other piezoelectric coefficients (d_{31} , d_{33} , and d_{14}) cannot be obtained because the electric potential cannot be applied along the fibril's longitudinal direction to measure d_{31} and d_{33} and a collagen fibril was fixed on the substrate resulting in nullification in the d_{14} direction.

[In main text p.14]

First, the cantilever was carefully calibrated in the lateral (in-plane) direction. We obtained the optical lever sensitivity (OLS) of the cantilever in the vertical direction from the force curves measured on a sapphire surface. The vertical OLS (V/nm) is a ratio of the sensing voltage of an AFM photodetector to the vertical deflection of the cantilever. Then, the lateral OLS can be estimated by the geometrical relationship between the vertical OLS and the lateral OLS as defined by^{65-67,78}.

$$R = \frac{\text{VinvOLS}}{\text{LinvOLS}} = \frac{\frac{\Delta z}{\Delta D_V}}{\frac{\Delta d}{\Delta D_L}} = \frac{\frac{L}{3S}}{\frac{h}{2S}} = \frac{2}{3} \cdot \frac{L}{h} \quad \text{Eqn. 3}$$

where Δz and Δd are out-of-plane and in-plane cantilever movements; ΔD_V and ΔD_L are the vertical and lateral movements of the laser spot on the photodiode; L is the cantilever length, S is the distance between the cantilever and the photodiode, and h is the height of the tip. R shows the

ratio between the inverse OLS in the vertical (VinvOLS, nm/V) and lateral (LinvOLS, nm/V) directions for a convenient calculation. In this study, the same cantilever was employed to measure both WT and OI bone samples, and the VinvOLS was 172.94 nm/V, and R was 23.81 based on the probe geometry of L=500 μm and h=14 μm. Thus, the LinvOLS was determined to be 7.31 nm/V.

In addition, we added details of how we got the quantitative piezoresponse in the main text and Fig. 7b and c.

[In main text p.16]

Finally, the actual piezoelectric strain (a_{piezo}) can be quantitatively determined by

$$a_{piezo} = \frac{A_{mv}}{Q} \cdot \text{LinvOLS} = \frac{A_{mv}}{Q} \cdot \frac{\text{VinvOLS} \cdot 3h}{2L} \quad \text{Eqn. 5}$$

where A_{mv} is the measured PFM amplitude in millivolt at the contact resonance frequency. For each PFM measurement, the contact resonance curve was obtained and fitted by Eqn. 1 to find the contact resonance frequency and Q factor. Figures 7b and c show a representative result of the contact resonance and fitted curve of the first lateral mode measured on WT and OI collagen, respectively. From the fitted curves, the contact resonance frequency ($\omega_0/2\pi$) and quality factor (Q) were obtained to be 251.5 kHz and 130.3 in the WT collagen and 236.1 kHz and 53.3 in the OI collagen.

Figure 7(b),(c)

Reviewer #3 (Remarks to the Author):

The manuscript presents a piezoresponse AFM study to measure at high-spatial resolution the piezoelectricity of collagen fibers. Specifically, the manuscript compares the piezoelectrical response between gap and overlap regions of wild-type (WT) collagen fibers from collagen fibers obtained from osteogenesis imperfecta (OI) bones. The measurements reveal little differences between the piezoelectrical response of gap and overlap regions in OI collagen. That results is in contrast with the behaviour observed on WT collagen. The authors interpret the results as a consequence of the mineralization of collagen by calcium phosphate. More specifically, the manuscript claims a physiological role for collagen piezoelectricity.

This is an attractive, original and relevant contribution. The novelty of the results makes this manuscript suitable for publication in Communications Biology after some revision.

1 It is not very clear, but the experiments were performed in air. It is well-established that the nanomechanical properties of collagen fibers do depend on the environment (buffer solution, air...). The manuscript should acknowledge it.

Reply to comment 1. We truly agree with this comment. All PFM experiments were conducted in air and well-controlled laboratory. We included this information in our main text as shown below. We also added much more details about the experiment in the method section. Please refer to the method section in the revised manuscript.

[In the main text p.12]

We kept the samples in the buffer solution and made it dried for 30 minutes before the measurement. Also, the lab was maintained at the relative humidity of around 40% and the temperature of 20~23 °C.

2 At its best, the manuscript shows a correlation between piezoelectricity and mineralization. However, this correlation is also observed in the stiffness measurements (Fig. 4). In my view, the available experimental data does not support a physiological relevance for the piezoelectrical properties of collagen.

Reply to comment 2. We appreciate this comment. The correlation shown in the stiffness measurements infers that OI bone experiences abnormal mineralization. However, the underlying mechanism for the abnormal mineralization cannot be explained.

In this work, we hypothesized that the collagen piezoelectricity is the mechanism functioning as a stiffness modulator during the intrafibrillar mineralization process. The higher piezoelectricity at the gap region in the WT de-mineralized collagen helps the ACP mineral precursors infiltrate into the gap zone. Interestingly, however, the de-mineralized collagen of OI bone loses its piezoelectric periodicity and heterogeneity along the collagen fibril. This corroborates our hypothesis in that losing piezoelectric heterogeneity (or periodicity) results in losing heterogeneity (or periodicity) in the mechanical property, inferring that piezoelectricity is the underlying mechanism for the abnormal mineralization

in OI bone. Thus, we could claim that these PFM results suggest an important physiological role of the collagen piezoelectricity in the intrafibrillar mineralization.

3 The units were missing in Fig. 4.

Reply to comment 3. Thanks for the comment. Since Fig. 4 is not actual data, we intentionally did not include the units. Fig. 4 is a schematic cartoon image to explain the relationship between topography, piezoresponse, and stiffness of the WT and OI collagen.

4 To set the current scientific context on collagen characterization at the nanoscale, the authors might consider to cite a recent contribution on the nanomechanical properties of collagen nanoribbons: Gisbert, V. G. et al. ACS Nano 15, 1850-1857 (2021).

Reply to comment 4. Thanks for introducing the reference. We included this paper in ref (50).

Reviewers' comments:

Reviewer #1 (Remarks to the Author):

The authors addressed most of the concerns I had regarding their methodology with their edits to the manuscript and rebuttal document. However, aspects of the response still seem to indicate a miss interpretation of some parts of the analysis of the PFM signal. I will leave it to the editor to determine if these details warrant an additional round of revisions.

1. In response to reviewer 1, Comment 2: The authors state "we believe that the shape of the laser spot does not affect the sensitivity unless the laser shape is asymmetric."

I agree with this statement. The issue is that the Asylum Research MFP3d AFM system has an "asymmetric" laser spot. The spot is longer (along the length of the cantilever) than it is wide. An odd detail here is the specific AFM system used in the study is specified as an Asylum Research MFP3d AFM in the previously submitted version of the manuscript (line 325). However, I could not find a statement indicating which AFM system was used in the current manuscript.

2. In response to reviewer 1, Comment 3: The authors state that "These [PFM] point measurement results are not affected by the variations of height and material properties, because each data point is calibrated to eliminate such artifacts."

I disagree with this statement. If the DART tracking is behaving perfectly, the mechanical properties of the sample identically affect observed amplitude and resonance frequency in both DART and point measurements (I am unsure what "calibration" the authors are referring to here). A decrease in sample stiffness would both decrease the resonance frequency and decrease the observed amplitude. In my opinion, the fundamental question is: "Is the change in amplitude due to stiffness significant when compared to the magnitude of the PFM amplitude signal." The other part of the author's response has shown that the frequency shift in question is small. My general concern here has been adequately addressed.

Reviewer #2 (Remarks to the Author):

The authors had done a lot of revisions to address the technical aspects of the PFM testing. However, I am still not convinced by their interpretation of the PFM data as being due to the higher piezoelectric activity of the gap regions (Fig 2 and 7). Even if we ignore a difference in the resonance frequency between overlap and gap regions due to different stiffness (BTW, can we exclude a presence of the residual mineral phase in the gap regions in the demineralized samples?), there are quite large topographic variations of 1-2 nm which are enough to cause additional artifact related to the capacitance effect. In weak piezoelectrics, like collagen, this effect could be more profound than a genuine piezoelectric signal. Did the authors try the out-of-plane PFM imaging? I expect that it will also produce an enhanced signal at the gap regions, which would be difficult to attribute to the enhanced piezoelectricity as d_{33} value is an order of magnitude lower than d_{15} . There is still no information on the sample thickness, type of the conducting substrate.

Reviewer #3 (Remarks to the Author):

The authors have satisfactorily addressed my comments. The revised version is suitable for publication.

We would like to sincerely thank the reviewers for their comments. Our responses to reviewers' comments are given by the following notation:

- *reprinted comments from the reviewer written in italics*
- **our response written in blue**
- **revision in the paper highlighted by yellow.**

Reviewers' comments:

Reviewer #1 (Remarks to the Author):

The authors addressed most of the concerns I had regarding their methodology with their edits to the manuscript and rebuttal document. However, aspects of the response still seem to indicate a miss interpretation of some parts of the analysis of the PFM signal. I will leave it to the editor to determine if these details warrant an additional round of revisions.

1. In response to reviewer 1, Comment 2: The authors state “we believe that the shape of the laser spot does not affect the sensitivity unless the laser shape is asymmetric.”

I agree with this statement. The issue is that the Asylum Research MFP3d AFM system has an “asymmetric” laser spot. The spot is longer (along the length of the cantilever) than it is wide. An odd detail here is the specific AFM system used in the study is specified as an Asylum Research MFP3d AFM in the previously submitted version of the manuscript (line 325). However, I could not find a statement indicating which AFM system was used in the current manuscript.

Thanks for the comment. We used MFP3D AFM from Asylum research. During the revision, we somehow unintentionally missed the information. We added this information back in the manuscript as shown below (page 14).

In this study, the PFM was conducted with a commercial AFM system (MFP-3D Infinity, Asylum Research) using a conductive cantilever (2.5 N/m of nominal stiffness, 3XC-GG, OPUS®).

As the reviewer agreed, a symmetric laser spot does not affect the sensitivity. The MFP3D laser has an ellipse shape, longer in the axis of the cantilever-length, but **it is still symmetric about cantilever-lengthwise axis and about the crosswise axis**. As far as the shape is symmetric, the difference in length does not cause misinterpretation. There are many published papers using the same method for the MFP3D system as shown below:

1. Denning, D. *et al.* Piezoelectric Tensor of Collagen Fibrils Determined at the Nanoscale. *ACS Biomater. Sci. Eng.* **3**, 929–935 (2017).
2. Peter, F., Rudiger, A., Szot, K., Waser, R. & Reichenberg, B. Sample-tip interaction of piezoresponse force microscopy in ferroelectric nanostructures. *IEEE Trans. Ultrason. Ferroelectr. Freq. Control* **53**, 2253–2260 (2006).

2. In response to reviewer 1, Comment 3: The authors state that “These [PFM] point measurement results are not affected by the variations of height and material properties, because each data point is calibrated to eliminate such artifacts.”

I disagree with this statement. If the DART tracking is behaving perfectly, the mechanical properties of the sample identically affect observed amplitude and resonance frequency in both DART and point measurements (I am unsure what “calibration” the authors are referring to here). A decrease in sample stiffness would both decrease the resonance frequency and decrease the observed amplitude. In my opinion, the fundamental question is: “Is the change in amplitude due to stiffness significant when compared to the magnitude of the PFM amplitude signal.” The other part of the author’s response has shown that the frequency shift in question is small. My general concern here has been adequately addressed.

We agree that the resonance frequency is affected and shifted by the mechanical properties, and the amplitude is changed accordingly. To compensate these effects, **we obtained each frequency response curve at each point to identify the individual resonance peak.** For each curve, we obtained the resonance frequency (ω_o) and Q factor to deduce the amplitude of piezoreponse, using Eqn. 1.

$$A(\omega) = \frac{\alpha_{piezo}}{\sqrt{\left(1 - \left(\frac{\omega}{\omega_0}\right)^2\right)^2 + \left(\frac{\omega}{Q \cdot \omega_0}\right)^2}} \quad \text{Eqn. 1}$$

$$\tan\phi(\omega) = \frac{\omega/\omega_0}{Q\left(1 - \left(\frac{\omega}{\omega_0}\right)^2\right)}$$

where ω and ω_0 are the drive frequency and contact resonance frequency, respectively, α_{piezo} is the piezoresponse amplitude, and Q is the quality factor of the contact resonance.

Note that we calculated the piezoresponse value based on the peak value (i.e., $\omega = \omega_0$) in each individual resonance frequency curve (i.e., $\alpha_{piezo} = A(\omega_o)/Q$). Therefore, **the effect of frequency shift was minimized in our point-measurements. The effect of stiffness is intrinsically included in each contact resonant curve.** We fully described detailed information and procedure of the point-measurement in the section on quantitative PFM calibration (page. 16) and the previous rebuttal.

We apologize that we did not clearly mention about the calibration in the rebuttal, while we included all the detailed information in the manuscript (page. 16). To be more precise, the statement should be changed to “For every PFM point measurement, the change in the resonance frequency is compensated by measuring the resonance curve at each measurement. Thus, the artifact due to the frequency shift caused by material property changes was minimized in the calibrated results of point measurements.”

Additionally, DART-PFM results are supportive data. We agree that DART-PFM is not a perfect method to remove all crosstalk effects. So, if the DART-PFM results were different from the point-measurement results, it would indicate the existence of the crosstalk effect. However, the DART-PFM data, fortunately, follow the same trend as the point-measurement results, indicating no such significant crosstalk, and this result does not undermine the novelty of this work. Also, as pointed by the reviewer, we provided the data showing the frequency shift during

DART-PFM is small. **Therefore, we are highly confident that all concerns of the methodologies have been properly addressed.**

Reviewer #2 (Remarks to the Author):

The authors had done a lot of revisions to address the technical aspects of the PFM testing. However, I am still not convinced by their interpretation of the PFM data as being due to the higher piezoelectric activity of the gap regions (Fig 2 and 7). Even if we ignore a difference in the resonance frequency between overlap and gap regions due to different stiffness (BTW, can we exclude a presence of the residual mineral phase in the gap regions in the demineralized samples?),

About demineralization: Fully mineralized collagen fibrils take a few hours to remove the all residual mineral phase inside it (<https://doi.org/10.1016/j.jsb.2008.02.010>). Also, fully demineralization of mouse teeth which is the highest mineralized tissue takes a few days (<https://www.ncbi.nlm.nih.gov/pmc/articles/PMC3040998/>). Here, we demineralized the sample for 14 days (much longer than required several days), so we are confident that the sample was well demineralized. In our earlier publication, a type I collagen fibril, not demineralized collagen, (<https://doi.org/10.1021/acsbiomaterials.0c01314>) also exhibited similar results as the demineralized collagen sample.

there are quite large topographic variations of 1-2 nm which are enough to cause additional artifact related to the capacitance effect. In weak piezoelectrics, like collagen, this effect could be more profound than a genuine piezoelectric signal.

We agree with the concern that the resonance frequency is affected and shifted by the mechanical properties, topography, or capacitance effect. However, we provided all details about a multi-pronged PFM approach to build credentials about our data. We fully described how we addressed the artifacts due to frequency shift related with topography and mechanical properties in the point-measurement. We also provided all the details about the capacitance effect. If you think that our supporting data is not enough, please guide us what else data should be provided.

We included the details about the capacitance effect in the earlier response, in addition to the out-of-plane PFM imaging. Out-of-plan (or vertical) PFM imaging is largely affected by the capacitance, but **the lateral PFM we are performing is not affected by the capacitance**. The answer we included in the previous revision is repeated below (see Reply to comment 2).

Even though there are several published papers showing lower piezoelectricity at the gap regions, these results were obtained by the conventional PFM method, which is highly vulnerable to mechanical stiffness changes. **The novelty of the current work is that we applied a multi-pronged PFM approach, based on point-measurements and DART-PFM, to unambiguously distinguish the piezoelectric heterogeneity.** We already published another paper showing the same trend, higher piezoelectricity at the gap zone, obtained from a single type I collagen fibril (<https://doi.org/10.1021/acsbiomaterials.0c01314>).

Did the authors try the out-of-plane PFM imaging? I expect that it will also produce an enhanced signal at the gap regions, which would be difficult to attribute to the enhanced piezoelectricity as d_{33} value is an order of magnitude lower than d_{15} .

Yes, we did. As shown in the previous rebuttal, we tried out-of-plane PFM imaging. After compensating the electrostatic effect on the out-of-plane PFM result by applying DC voltage offset, **we didn't get any significant signal from the out-of-plane PFM**. The below PFM scanning image shows the out-of-plane PFM result of a collagen fibril. To compensate for the capacitance effect, 180 mV [DC] was applied additionally with 1V [AC] drive amplitude. As seen in the figure, no significant piezoresponse was observed. The little variation on the collagen fibril and high response on the substrate is because of the local capacitance effect. Also, we described the details of the capacitance effect on the PFM result in the previous rebuttal, which is repeated below.

Figure R4: The out-of-plane PFM amplitude map of a collagen fibril

Reply to comment 2. We totally agree with this comment that the electrostatic contribution should be carefully addressed. In our previous work, we investigated the electrostatic contribution on PFM results, and we successfully demonstrated that the lateral PFM is not affected by the electrostatic force (Ref. 31). This is because the capacitance derivative term (dC/dZ) in the electrostatic equation is averaged out during the lateral PFM (Ref. 72).

$$A_{electrostatic} = \left| k^{-1} \left(\frac{dC}{dz} \right) V_{ac} (V_{dc} - V_{sp}) \right| \cdot Q$$

To examine and eliminate the electrostatic effect, we measure the PFM amplitude with varied DC voltage (V_{dc}) as shown in Figure R4 (brought from our previous paper, ref.31). In vertical (out-of-plane) PFM (cf. Fig. R4c left), a V-shape of PFM amplitude was revealed depending on V_{dc} because of the electrostatic interaction. In lateral (in-plane) PFM (cf. Fig. R4c right), however, the PFM amplitude was not varied by V_{dc} , indicating that lateral PFM is not affected by electrostatic interactions.

Figure R2: The effect of electrostatic forces on vertical PFM and lateral PFM

There is still no information on the sample thickness, type of the conducting substrate.

We apologize that we missed this in the earlier revision. The information of substrate was included in the main text, and we added the sample thickness information at page 14 as shown below. The presented PFM amplitude is based on strain to voltage, not strain to electric field, and thus the thickness information will not make any change in the result.

The prepared midshaft bone samples have 3~4 μm thickness.